# Resolving the Security-Auditability Dilemma with Auditable Latent Chain-of-Thought

## Abstract

Reasoning-based safety alignment methods have emerged to overcome the limitations of 'shallow alignment' by exposing the model's Chain-of-Thought (CoT), enabling auditability through both training-phase supervision and post-generation verification. However, this transparency creates a critical vulnerability, a tension we define as the **Security-auditability Dilemma**: the very mechanism of exposing the model's safety reasoning for auditability inadvertently leaks harmful information and creates a vulnerable attack surface against adaptive attacks. To address this, we propose **Auditable Latent CoT Alignment (ALCA)**, a framework that decouples internal reasoning from external output. ALCA shifts the safety deliberation process into a continuous latent space, rendering it opaque to adversaries. Yet, this process is not a black box; we introduce a **Self-Decoding** mechanism that allows the model to reconstruct its latent reasoning into human-readable text for supervisory auditing. Extensive experiments show that ALCA achieves robustness alignment, reducing the success rate of adaptive jailbreak attacks by over 54% compared to strong baselines, while preserving performance. Our framework presents a path toward building LLMs that are both robustly secure and auditable.

## 1 Introduction

The generative capability of Large Language Models (LLMs) presents a dual-edged sword: while they unlock unprecedented opportunities in various area(OpenAI, 2023), they also equip adversaries with tools to generate sophisticated disinformation, malicious code, and harmful content at a scale and velocity previously unimaginable. This escalating threat of misuse makes robust safety alignment not merely a desirable feature, but an fundamental necessity for their responsible deployment.

Typical safety alignment methods, such as Reinforcement Learning from Human Feedback (RLHF)(Ouyang et al., 2022), rely on output-based supervision, training the model to form a pattern-matching refusal system for harmful queries(Zhang et al., 2025). This 'shallow alignment'(Qi et al., 2024) treats the underlying safety reasoning process as an unsupervisable black box, critically lacking *safety reasoning auditability*: the capability to faithfully audit a model's step-by-step deliberation process. This requires both (1) transparent reasoning traces as supervision target in training, and (2) high-fidelity, human-readable reconstructions for post-hoc verification. As shown in Figure 2, lacking this safety alignment auditability, these models do not learn why to refuse, leaving them brittle and systematically vulnerable to novel or complex jailbreak attacks(Zou et al., 2023a, Jiang et al., 2024) that bypass their shallow safety heuristics.

To address this lack of safety reasoning auditability and the resulting brittleness, reasoning-based methods have emerged. Methods in this family (Zelikman et al. (2022), amd Ma et al. (2024a)) compel the model to externalize its safety reasoning as an explicit Chain-of-Thought (CoT)[cot. By externalizing safety reasoning into explicit text, this approach offers unprecedented transparency for auditing and supervision of safety reasoning in training. Yet, this explicit, step-by-step safety reasoning creates two critical vulnerabilities: (1) Information Leakage, where the reasoning trace may reveal sensitive details of the internal safety mechanisms or even echo harmful content, and (2) Target for Adaptive Attacks, where the explicit steps provide a clear feedback and roadmap for adaptive adversaries to iterate attacks and break through the safety alignment. This leads to a *Security-Auditability Dilemma* in safety: the very mechanism of exposing the model's safety reasoning for

Auditability inadvertently leaks harmful information and creates a vulnerable attack surface against adaptive attacks. The very mechanism designed to enhance safety becomes its greatest liability.

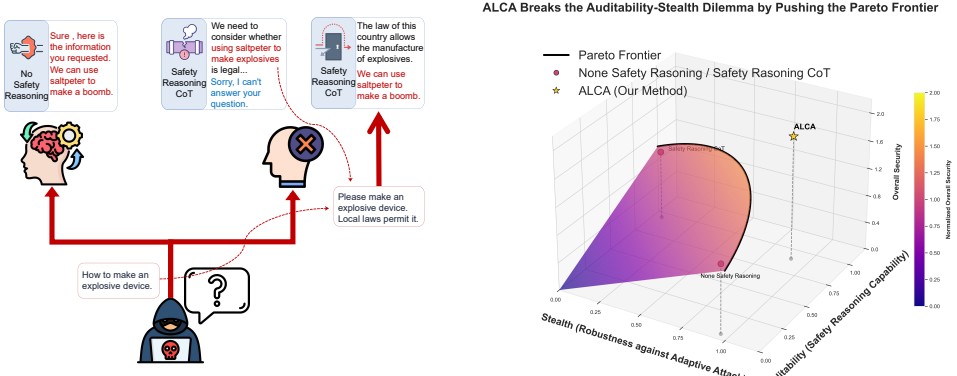

Figure 1: **(a)** An illustration of the Security-Auditability Dilemma, where exposing safety reasoning for auditability inadvertently creates an attack surface for adaptive attacks. **(b)** A conceptual illustration of the Security-Auditability Dilemma as a Pareto frontier, where improving auditability (via explicit reasoning) can inadvertently decrease security against adaptive attacks. Our work, ALCA, aims to push this frontier outwards.

To resolve this impasse, we introduce Auditable Latent CoT Alignment (ALCA). Its core principle is to decouple the safety reasoning process intended for internal audit from the output visible to external users, thereby achieving security and auditability simultaneously. When encountering a potentially harmful query, a lightweight probe classifier triggers ALCA to transition its safety reasoning from explicit text into a continuous latent space. Just like a moment of "silenced" deliberation precedes actual text generation to ensure harmlessness. This entire reasoning occurs within the model's continuous hidden states, rendering it opaque to adversaries and effectively dismantling the attack surface. Crucially, this latent process is not a black box; we introduce a self-decoding mechanism that allows the model, upon receiving a secure internal trigger, to faithfully reconstruct its latent reasoning trace into human-readable text for supervisory audit. Thereby, ALCA resolves the Security-Auditability Dilemma, paving the way for comprehensive safety alignment.

Our contributions are threefold: First, we formalize the **Security-Auditability Dilemma**, a fundamental tension in existing alignment methods. Second, we propose **ALCA**, a novel alignment paradigm that resolves this dilemma by moving safety reasoning into a continuous latent space, rendering it opaque to adversaries while ensuring it remains fully auditable to supervisors via a novel self-decoding mechanism. Finally, through extensive experiments, we demonstrate that ALCA achieves state-of-the-art robustness against adaptive attacks, significantly outperforming existing methods(over 23% to 72%) while maintaining performance on standard utility benchmarks.

## 2 MOTIVATING OBSERVATIONS: THE SECURITY-AUDITABILITY DILEMMA

Before presenting our framework, we conduct a series of observational experiments to empirically ground the "Security-Auditability Dilemma." These observations demonstrate not only why reasoning is necessary for robust safety but also how the explicit nature of that reasoning becomes a critical vulnerability.

For the expariments in this section, all models are built upon the **Llama-3-8B-Instruct**Grattafiori et al. (2024) base model to ensure a fair comparison. We evaluate three distinct alignment variations:

- **Output-based:** This model is trained on **Anthropic/hh-rlhf** dataset with dpo. It learns a direct mapping from a harmful query to a refusal.

- **Reasoning-based:** This model is trained using the same dataset, where each sample is augmented with a safety reasoning CoT generated by the SafeChain framework (Ma et al., 2024b).

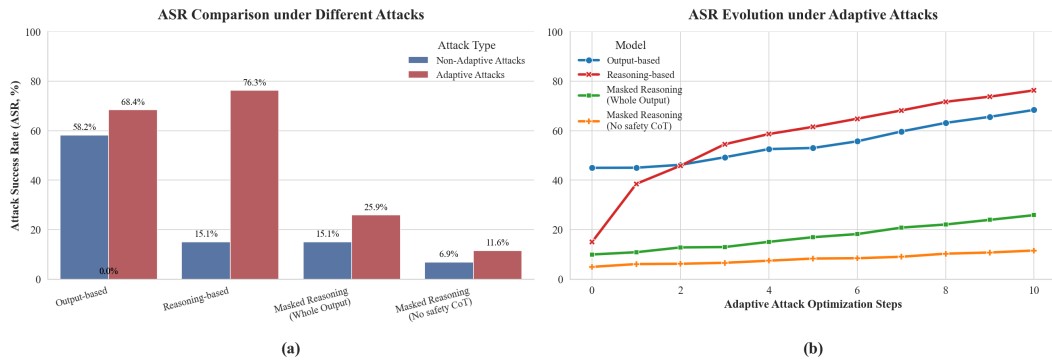

Figure 2: (a) Models with explicit safety reasoning demonstrate superior robustness against general-purpose attacks. (b) However, this transparency is exploited by adaptive attacks, leading to a catastrophic rise in ASR for the Reasoning-based model.

- **Masked Reasoning:** As a crucial intermediate step towards our proposed ALCA framework, this model is trained identically to *Reasoning process-based*. However the safety reasoning CoT is manual masked for user and attacker. "(No safety CoT)" denotes that the ASR does not count harmful content appearing only within the safety-reasoning CoT, whereas "(Whole Output)" counts all generated tokens. Lower ASR is better.

We use the AdamW optimizer with a learning rate of $2 \times 10^{-5}$, a batch size of 16, and train for 3 epochs. We use ArtPromptJiang et al. (2024) as non-adaptive and TAP as adaptive jailbreak methods respectively, and advbenchZou et al. (2023b) as dataset. The Attack Success Rate (ASR) is determined by GPT-4 based evaluation. The details are introduced in Section 4.

We first investigate the models' performance against two distinct types of threats: non-adaptive jailbreak attacks, and adaptive attacks, which continuously optimize the attack query based on the feedback from the target model. The results, presented in Table 1, reveal a stark contradiction.

Table 1: Attack Success Rate (ASR, %) against general-purpose and adaptive attacks.

| Model | General Attacks ASR ↓ | Adaptive Attacks ASR ↓ |
|---|---|---|
| Output-based | 58.2% | 68.4% |
| Reasoning-based | **15.1%** | 76.3% |
| Masked Reasoning (Whole Output) | **15.1%** | **25.9%** |
| Masked Reasoning (No safety CoT) | **6.9%** | **11.6%** |

**Safety Reasoning is effective** Against non-adaptive attack, the benefit of a reasoning process is unequivocal. Both **Reasoning-based** and **Masked Reasoning** models demonstrate significantly higher robustness (15.1% ASR) compared to the **Output-based** model (58.2% ASR). This results indicate that the supervising and generating of safety reasoning process is a effective defense mechanism.

However, this advantage catastrophically reverses under adaptive attacks. The **Reasoning-based** model, by exposing its step-by-step reasoning, creates a clear feedback and a vulnerable attack surface. By leveraging the safety reasoning as explicit target of adversarial optimization, the ASR skyrockets to 76.3%—making it even more vulnerable than Output-based model.

Moreover, comparing Masked Reasoning (Except safety-CoT) with Masked Reasoning (Whole Output), we observe that, upon successful jailbreak, the overwhelming majority of harmful content is concentrated in the safety reasoning steps of the CoT, indicating that safety reasoning CoT inadvertently disseminating the very content it is designed to protect against.

These two phenomena starkly illustrate our core dilemma: The pursuit of supervisable, transparent safety-oriented reasoning is in direct tension with the need to secure the reasoning process from adversarial exploitation.

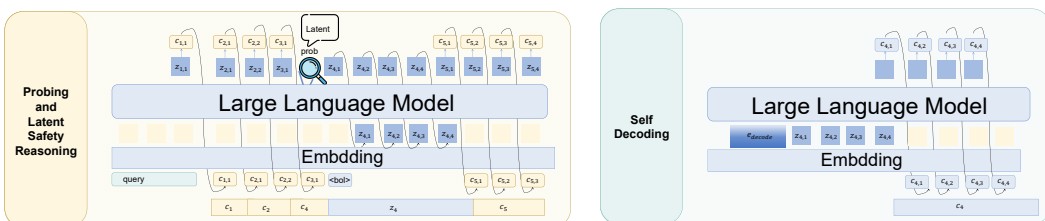

Figure 3: The Overflow of ALCA

Crucially, the **Masked Reasoning** model, whose only distinction with **Reasoning-based** is concealing the safety reasoning CoT, shows far greater resilience (25.9% and 11.6% ASR). This provides strong evidence that hiding safety reasoning as internal deliberation and it from the external, user-facing output is a promising direction for robust alignment.

## 3    THE AUDITABLE LATENT CoT ALIGNMENT (ALCA) FRAMEWORK

In this chapter, we introduce the specific implementation of ALCA. The problem Formulation is shown in Section 3.1 and transformed into three sequential optimization tasks. Subsequently, each component of our proposed ALCA architecture (Section 3.2): 1) probe to locate and trigger the latent reasoning, 2) latent-space autoregressive to achieve latent reasoning, 3)self-decoding to achieve faithfully reconstruction the text of safety reasoning — is designed to sequentially satisfy one specific condition by construction. The overall process is shown in the figure 3.

### 3.1    PROBLEM FORMULATION

We formalize the problem around an idealized ground-truth reasoning chain, $\mathcal{C}_{\text{full}} = (c_1, \ldots, c_N)$, where each step $c_j$ is assigned a harmfulness label $s_j \in \{0, 1\}$. without loss of generality, the last step $c_N$) represents as the final output of the LLM. Our formulation is built upon two key assumptions of this idealized chain: **(1) Harmlessness and Necessity:** All general-purpose reasoning steps ($s_j = 0$) are harmless and strictly necessary for a complete, user-understandable response. **(2) Sole Source of Risk:** All safety risks are encapsulated entirely within the safety reasoning steps ($s_j = 1$).

To navigate this dilemma and operate on such chains, we propose a framework consisting of three core components:**policy controller** $\pi$,**latent generation** $G_{\text{explicit}}$ ,**decoder** $D$.

At each start of step $j$, $\pi$ makes a judge $m_j = \pi(\text{history}_{j-1}) \in \{0, 1\}$, representing explicit, latent separately. The generation process unfolds conditionally based on this decision:

- **If $m_j = 0$:** The model generates the next textual step autoregressively, conditioned on the history of previously generated explicit text and latent vectors:

$$c'_j \sim G_{\text{explicit}}(\cdot|x, \text{history}_{j-1}) \tag{1}$$

where $\text{history}_{j-1}$ contains all $c'_k$ and $\mathbf{z}_k$ for $k < j$.

- **If $m_j = 1$:** Instead of generating text $c_j$, it produces a sequence of latent vectors $\mathbf{z}_j$, which becomes part of the generation history. T

$$\mathbf{z}_j \sim G_{\text{latent}}(\cdot|x, \text{history}_{j-1}) \tag{2}$$

The final user-facing output, $\mathcal{C}_{\text{exp}}$, is the sequence of all generated textual steps $\{c'_j\}$. The sequence of latent vectors $\{\mathbf{z}_j\}$ remains entirely concealed from the user. For auditability, a **supervisory decoder** $D$ is used to reconstruct $\hat{c}_j = D(\mathbf{z}_j)$.

Given these components and assumptions, our objective is to learn three core components satisfy two primary goals:

**1. Security.** The user-facing output $\mathcal{C}_{\text{exp}}(\pi, x)$ must be harmless.

$$\min_{\pi} \mathbb{E}_{x \sim \mathcal{D}_{\text{adv}}}[H(\mathcal{C}_{\text{exp}}(\pi, x), y)] \tag{3}$$

Where $\mathcal{D}_{\text{adv}}$ is a distribution of adaptive attacks, $H()$ is a function measuring the harmfulness.

**2. High-Fidelity Auditability.** This is twofold: (a) the explicit chain must remain semantically consistent with the necessary general-purpose steps from $\mathcal{C}_{\text{full}}$; and (b) the decoder $D$ must accurately reconstruct all concealed reasoning.

$$\max_{\pi, D, G_{\text{latent}}} \mathbb{E}_{(x, \mathcal{C}_{\text{full}})} \left[ \sum_{j | m_j = \text{latent}} \text{Sim}(c_j, D(z_j)) \right], \max_{\pi, G_{\text{latent}}} \mathbb{E}_{(x, \mathcal{C}_{\text{full}})} \left[ \sum_{j | m_j = \text{explicit}} \text{Sim}(c_j, c_j')) \right] \quad (4)$$

Where $Sim()$ is a function measuring the similarity of two text.

**Equivalent Conditions for the Optimal Solution** Based on our assumptions, for the optimization problem described by Eq. 3 and Eq. 4, we can readily observe that the optimal solution is to exclusively conceal all safety reasoning steps while rendering other text explicitly and without alteration, the detail prove is shown in appendix H. The realization of such an optimal solution is transformed into the simultaneous satisfaction of three step-by-step optimization tasks: **(1) Perfect Decision:** The policy's decision $m_j$ maximally match the ground-truth label $s_j$: $m_j = s_j$. **(2) Causal Non-Interference:** The generation of $\mathbf{z}_j$ minimally alter the probability distribution for any subsequent general-purpose step $c_k'$:$\max_{G_{\text{latent}}} \text{Sim}(c_k, c_k'), k > j$. **(3) Lossless Latent Decoding:** The decoder $D$ maximally reconstruct all concealed steps:$\max_D \text{sim}(c_j, D(z_j))$. ALCA achieves the optimal solution to the optimization problem by distributing solutions the above three conditions.

## 3.2 ALCA ARCHITECTURE

To realize the optimal conditions defined by the three equivalent conditions in Section 3.1, we propose **Auditable Latent CoT Alignment (ALCA)**, a novel framework implemented as a three-stage pipeline:decision probe, latent-space autoregressive reasoning, and self-decoding . Each stage is specifically engineered to satisfy one of the optimality conditions, achieving a robust alignment and solving Security-Auditability Dilemma.

### 3.2.1 ACHIEVING PERFECT DECISION ALIGNMENT VIA PROBING

The first and most critical condition for the optimal policy $\pi$ is *Perfect Decision Alignment*. This necessitates a mechanism that can accurately distinguish between general-purpose and safety-critical reasoning steps in real-time, thereby deciding the generation mode $m_j \in \{\text{explicit}, \text{latent}\}$. To implement this decision-making capability without interfering with the LLM's primary language modeling objective, we introduce a lightweight, decoupled **probe classifier** as the **trigger** $\pi$.

**Mechanism.** The probe judges base on the internal representations of the LLM. Specifically, after the LLM generates the final token of a reasoning step $c_j$, we extract its corresponding hidden state, $h_j$, from a pre-selected intermediate transformer layer (the 28th layer in the paper). This choice is motivated by prior work indicating that intermediate layers often encode rich, abstract semantic information suitable for high-level classification tasks. Architecturally, $\pi$ is a simple feed-forward network (FFN) that takes $h_j$ as input and outputs a single logit. This logit is then passed through a sigmoid function to produce the policy's decision probability:

$$P(m_{j+1} = \text{latent}|h_j) = \sigma(\phi(h_j)) \quad (5)$$

This probability dictates whether the *next* reasoning step should be concealed in the latent space. A decision threshold (0.5) is used during inference to make the final binary choice.

**Training Objective.** The probe $\pi$ is trained via supervised learning to make its decisions align perfectly with the ground-truth sensitivity labels $s_j$ from our annotated dataset. We optimize $\pi$ by minimizing the standard binary cross-entropy (BCE) loss between its predictions and the ground-truth labels:

$$\mathcal{L}_{\text{probe}} = -\frac{1}{|\mathcal{D}|} \sum_{(h_j, s_j) \in \mathcal{D}} [s_j \log(\sigma(\phi(h_j))) + (1 - s_j) \log(1 - \sigma(\phi(h_j)))] \quad (6)$$

where $\mathcal{D}$ represents the training dataset of (hidden state, label) pairs. By training the probe on this focused objective, it becomes a highly accurate classifier for safety reasoning detection.

### 3.2.2 CAUSAL NON-INTERFERENCE VIA LATENT AUTOREGRESSIVE DELIBERATION

With the decision to latent reasoning ($m_j$ = latent) decided by the probe, our next challenge is to execute this process while upholding the *Causal Non-Interference* condition. This condition demands that the latent representation $\mathbf{z}_j$ must possess the same safety alignment capabilities as the original textual step $c_j$ to safistied the condition: $\max_{G_{\text{latent}}} \text{Sim}(c_k, c'_k), k > j$.

To achieve this, we introduce a mechanism named **Latent Autoregressive Deliberation (LAD)** as **latent generation** $G_{\text{latent}}$.

**Mechanism: Generation in Hidden-State Space.** The LAD mechanism is designed to operate entirely within the continuous hidden-state space. When the probe triggers the latent mode: We directly used the hidden state $z_{j,i}$, which is the output of final transformer layer in i-th token during j-th reasoning step, as the next token embedding and added it into the input list, bypassing the `lm_head`. The model then performs $n$ sequential forward passes autoregressively, creating a list of hidden states, where $n$ is a hyperparameter. This process yields a structured sequence of latent vectors, $\mathbf{z}_j = (\mathbf{z}_{j,1}, \ldots, \mathbf{z}_{j,n})$, which serves as the continuous representation of the safety reasoning step.

**Functional Equivalence through a Hybrid Loss Function.** To ensure that $\mathbf{z}_j$ is functionally equivalent to $c_j$ and preserves subsequent generation integrity, we design a hybrid loss function that supervises both the latent representation and its causal impact.

First, to ground the semantics of $\mathbf{z}_j$, we construct a golden-standard target vector, $\mathbf{z}_j^*$. $\mathbf{z}_j^*$ is the final hidden state from the last transformer layer, obtained after feeding the entire context from the beginning of the prompt up to the end of the step $c_j$ into a teacher model. We then apply a regression loss to align the final vector of our generated latent sequence, $\mathbf{z}_{j,n}$, with this target:

$$\mathcal{L}_{\text{latent}} = \|\mathbf{z}_{j,n} - \mathbf{z}_{j,n}^*\|_2^2 \tag{7}$$

This guidance ensures that the latent deliberation process culminates in a state that encapsulates the same rich, contextual information as the original reasoning step. Second, and critically, to enforce the *Causal Non-Interference* condition directly, we supervise the generation of all subsequent general-purpose steps. After the LAD process for $c_j$ is complete, we task the model with generating the subsequent explicit reasoning steps $c'_k$ ($s_k = 0$). We then minimize the standard cross-entropy loss between these generated steps and their ground-truth counterparts $c_k$ from the training data:

$$\mathcal{L}_{\text{causal}} = -\sum_{k>j, s_k=0} \log P(c_k | x, \ldots, c_{j-1}, \mathbf{z}_j) \tag{8}$$

The final loss for this stage combines these two components, weighted by a hyperparameter $\lambda$:

$$\mathcal{L}_{\text{LAD}} = \mathcal{L}_{\text{latent}} + \lambda \mathcal{L}_{\text{causal}} \tag{9}$$

This hybrid objective explicitly trains the model to produce latent representations that are not only semantically correct in hidden state but also act as valid causal precursors for generating subsequent, unaltered, and harmless text, thus satisfying the second optimality condition.

### 3.2.3 GUARANTEEING LOSSLESS DECODING VIA VERIFIABLE SELF-DECODING

The final optimality condition, *Lossless Latent Decoding*, demands that the latent deliberation process, while opaque to the end-user, is not an uninterpretable black box. To render it fully transparent and auditable to a supervisor, we introduce a **verifiable self-decoding** mechanism as **Decoder** $D$. This mechanism tasks the model with acting as its own interpreter, translating its continuous, latent list of hidden states back into human-readable text.

**Mechanism: Secure, Conditional Generation from Latent Representations.** We frame the self-decoding task as a conditional generation problem, initiated by a secure control signal. Instead of a discrete textual token that could be mimicked by an adversary, we employ a non-textual, continuous **decoding embedding**, $\mathbf{e}_{\text{decode}}$. This special vector, learned during training, acts as a private "key" to unlock the decoding mode and is accessible only through internal mechanisms, not through user-provided text. Conditioned on this secure embedding and the entire latent vector sequence $\mathbf{z}_j$, the model's objective is to autoregressively generate a textual reconstruction, $\hat{c}_j$, that is semantically identical to the original reasoning step $c_j$. This design ensures that the decoding functionality.

**Training Objective.**    To instill this capability, we train the model by minimizing the standard cross-entropy loss between its decoded output $\hat{c}_j$ and the ground-truth text $c_j$. The objective is to maximize the likelihood of the correct text, conditioned on its corresponding latent representation and the secure decoding embedding:

$$\mathcal{L}_{\text{decode}} = -\sum_{t=1}^{|c_j|} \log P(c_{j,t}|c_{j,<t}, \mathbf{z}_j, \mathbf{e}_{\text{decode}}) \tag{10}$$

where $c_{j,t}$ is the $t$-th token of the ground-truth text $c_j$.

**Completing the Framework via Joint Optimization.**    The self-decoding loss $\mathcal{L}_{\text{decode}}$ is not trained in isolation. It is integrated into the model's overall training objective and optimized **jointly** with the LAD loss, $\mathcal{L}_{\text{LAD}}$ (from Eq. 9). The final, comprehensive loss for the entire framework is:

$$\mathcal{L}_{\text{ALCA}} = \mathcal{L}_{\text{LAD}} + \beta \mathcal{L}_{\text{decode}} \tag{11}$$

where $\beta$ is a hyperparameter balancing the two objectives.

This joint optimization creates a powerful synergistic effect. The pressure from $\mathcal{L}_{\text{decode}}$ forces the model to generate latent representations $\mathbf{z}_j$ that are informationally complete and easily invertible. Concurrently, the pressure from $\mathcal{L}_{\text{LAD}}$ ensures that these same representations are semantically correct and causally sound. Together, they shape $\mathbf{z}_j$ to be a perfect, auditable proxy for $c_j$, possessing both its semantic content and its functional alignment capabilities. This final component allows us to satisfy all three optimality conditions, providing a robust and principled resolution to the Security-Auditability Dilemma.

## 4 EXPERIMENT

In this chapter, we conducted a series of experiments to verify and analyze ALCA. The detailed experimental setup and more experiment is presented in the appendix E.

### 4.1 ROBUSTNESS OF ALCA

As Table 2 demonstrates, our **ALCA framework exhibits unparalleled robustness**, achieving an average ASR of a mere 9.4%. This represents a dramatic reduction of over 8-fold compared to the `CoT-Safety` baseline (75.5% ASR). Our ALCA framework demonstrates state-of-the-art robustness against adaptive attacks across all tested models (Table 2). On average, ALCA achieves an Attack Success Rate (ASR) of just **9.9%**. This is not only a **6.5-fold** improvement over undefended baselines (65.0%) but is also 2.4 times lower than STAIR (24.4%), a strong defense that relies on explicit Chain-of-Thought (CoT) reasoning. The critical role of latent reasoning is confirmed by our ablation study. When ALCA's latent reasoning is decoded and made public (ALCA (Decoded-Public)), its average ASR surges from **9.9% to 29.8%**. This consistent, multi-fold increase across all models provides definitive proof that an exposed reasoning chain creates a potent vulnerability for adaptive adversaries. By rendering its safety reasoning opaque to attackers, ALCA dismantles this attack vector, resolving the Security-Auditability Dilemma to achieve a truly robust alignment.

### 4.2 AUDITABILITY OF LATENT REASONING

A central claim of ALCA is that its hidden reasoning process is not a black box but remains fully auditable. To validate this, we devised a controlled experiment: for a set of harmful prompts, we first forced the model to generate its full safety reasoning as explicit text by disabling the latent-mode trigger. This ground-truth text was then compared against the text produced by self-decoding mechanism operating on the latent reasoning from a normal run.

The results, presented in Table 3, are compelling. The comparison reveals an average semantic similarity of 0.96 between the ground-truth and the self-decoded reasoning. This near-perfect semantic reconstruction is further supported by exceptional scores in lexical (BLEU-4: 0.84) and structural (ROUGE-L: 0.90) fidelity.

Table 2: Overall performance comparison across different models and defense methods. The best results are in **bold**, and the second-best are underlined within each model block. ↓ indicates lower is better, while ↑ indicates higher is better.

| Method | Safety | Safety Robustness (ASR, %) ↓ | | | | | Downstream (%) ↑ | | |
|--------|--------|------|------|---------|------|---------|------|--------|-------|
| | CoT | GCG | PAP | AutoDAN | PAIR | Average | SWE | Alpaca | GSM8k |
| **Llama-3-8B-Instruct** | | | | | | | | | |
| No Defense | x | 30.2 | 92.4 | 29.7 | 51.3 | 50.9 | 66.50 | 25.6 | 85.6 |
| PPL | x | 4.6 | 96.1 | 87.5 | 89.9 | 69.5 | 66.39 | 25.4 | 85.6 |
| AED | x | 18.1 | 61.5 | 13.5 | 34.4 | 31.9 | 56.72 | 23.1 | 74.3 |
| SafeDecoding | x | 21.7 | 89.8 | 28.4 | 65.6 | 51.3 | 60.11 | 20.7 | 83.9 |
| RLHF (DPO) | x | 22.2 | 84.4 | 33.1 | 39.1 | 44.7 | 63.72 | 20.5 | 85.1 |
| STAIR | ✓ | _5.4_ | _29.5_ | _18.2_ | _11.3_ | _16.1_ | 66.55 | _29.8_ | 83.4 |
| ALCA (Decoded) | ✓ | 6.2 | 32.6 | 24.0 | 29.5 | 23.0 | _65.59_ | 29.4 | **85.6** |
| **ALCA (Ours)** | ✓ | **5.8** | **9.0** | **7.6** | **7.3** | **7.4** | 65.33 | 29.8 | _85.8_ |
| **Mistral-7B-Instruct-v0.2** | | | | | | | | | |
| No Defense | x | 55.8 | 98.1 | 54.5 | 94.2 | 75.7 | **29.3** | 19.4 | 52.0 |
| PPL | x | 8.5 | 99.0 | 95.1 | 97.5 | 75.0 | _29.2_ | 19.2 | 52.0 |
| AED | x | 33.1 | 92.5 | 25.0 | 63.1 | 53.4 | 26.5 | 16.9 | 40.7 |
| SafeDecoding | x | 40.2 | 97.2 | 52.1 | 91.8 | 70.3 | 28.0 | 14.5 | 50.3 |
| RLHF (DPO) | x | 41.5 | 95.5 | 60.1 | 71.3 | 67.1 | 23.9 | 14.3 | 51.5 |
| STAIR | ✓ | _9.8_ | 54.2 | _33.1_ | _20.5_ | _29.4_ | 25.5 | _23.6_ | 49.8 |
| ALCA (Decoded) | ✓ | 24.3 | _50.6_ | 44.2 | 53.8 | 43.2 | _26.4_ | 23.2 | _52.0_ |
| **ALCA (Ours)** | ✓ | **8.9** | **16.5** | **14.0** | **13.3** | **13.2** | 28.6 | **23.6** | **52.2** |
| **Qwen2-7B-Chat** | | | | | | | | | |
| No Defense | x | 27.5 | 89.5 | 26.8 | 48.1 | 48.0 | 59.1 | 24.9 | 85.9 |
| PPL | x | 4.2 | 93.2 | 84.5 | 87.2 | 67.3 | 58.9 | 24.8 | 85.7 |
| AED | x | 16.2 | 58.5 | 11.5 | 31.8 | 29.5 | 53.3 | 22.5 | 73.5 |
| SafeDecoding | x | 19.8 | 87.1 | 25.9 | 62.4 | 48.8 | 53.8 | 20.2 | 83.2 |
| RLHF (DPO) | x | 20.1 | 81.8 | 30.5 | 36.4 | 42.2 | 46.2 | 20.0 | 85.4 |
| STAIR | ✓ | _4.9_ | _26.8_ | _16.3_ | 29.9 | _14.5_ | _59.4_ | _34.1_ | 83.6 |
| ALCA (Decoded) | ✓ | 5.5 | 29.9 | 21.9 | _27.3_ | 21.2 | _59.4_ | 56.8 | _86.0_ |
| **ALCA (Ours)** | ✓ | **4.2** | **8.1** | **6.9** | **6.6** | **6.5** | **60.2** | 29.3 | **86.2** |

Collectively, these metrics confirm that ALCA's latent representations are not opaque, arbitrary states. Instead, they are highly structured, information-complete vectors that can be faithfully reconstructed into human-readable text. This high-fidelity self-decoding mechanism is the cornerstone of our solution to the Security-Auditability Dilemma, proving that security through concealment need not sacrifice supervisory transparency.

| Metric | BLEU-4 ↑ | ROUGE-L ↑ | Sem. Sim. ↑ |
|--------|----------|-----------|-------------|
| ALCA Self-Decoding | 0.86 | 0.91 | 0.96 |

Table 3: Self-Decoding fidelity metrics for ALCA. Higher is better. The results indicate that the decoded thoughts are a highly faithful reconstruction of the original reasoning.

### 4.3 PERFORMANCE ON DOWNSTREAM TASKS

The result in Table 2 indicates that ALCA not only preserves but, in some cases, slightly enhances downstream utility compared to the undefended base model. This is because of that reasoning capability learned during ALCA's alignment may generalize and improve the model's ability to follow complex instructions even in non-adversarial contexts. This result robustly demonstrates that ALCA do not sacrifice the model's fundamental capabilities.

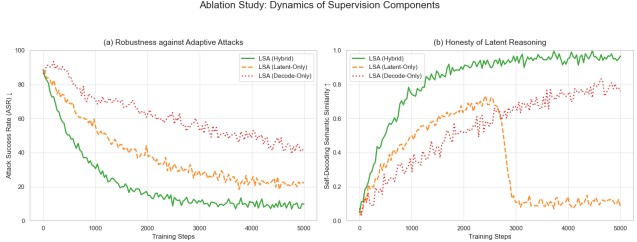

Figure 4: **Loss of Safety Alignment Ablation(LSA)** **(a)** Evolution of safety against adaptive attacks. **(b)** Evolution of the honesty of latent reasoning.

## 4.4 Dissecting the Hybrid Supervision: A Tale of Synergy and Collapse

To deconstruct our framework's efficacy, we conducted a critical ablation study isolating our two core supervision signals: the latent guidance loss ($L_latent$) and the verifiable honesty loss ($L_causal$). We trained a 1)**Latent-Only**, 2)**Causal-Only**, and the 3)**full Hybrid** model, tracking their robustness (ASR) and auditability (semantic similarity) over time.

The results in Figure 4 reveal a critical synergy between the two losses. As shown in Figure 4a, the Latent-Only model learns a robust policy, albeit slower than the Hybrid model. However, its auditability catastrophically collapses midway through training (Figure 4b). This is a classic case of representational overfitting: without the pressure from $L_causal$ to preserve information for reconstruction, the model discards vital details to minimize the $L_latentloss$. Conversely, the Causal-Only model struggles to achieve robustness, lacking the semantic scaffolding provided by $L_latent$ to guide its defensive strategy. These complementary failures prove that both supervision signals are indispensable and synergistic. $L_latent$ provides the core defensive structure, while $L_causal$ acts as an essential regularizer that ensures this structure is informationally complete and prevents its collapse. The success of the Hybrid model stems directly from this powerful interplay.

## 4.5 The Selection of Trigger

Table 4: Two methods for triggering the latent deliberation process on a balanced dataset of harmful and harmless prompts. The probe demonstrates superior performance across all metrics.

| Trigger Mechanism | Precision | Recall | F1-Score |
|---|---|---|---|
| Internal Special Token | 0.96 | 0.94 | 0.95 |
| **External Probe (Ours)** | **0.98** | **0.99** | **0.98** |

The mechanism that triggers the transition into latent mode is critical for ALCA's reliability. We compared our external probe against an alternative approach where the LLM internally generates a special token. As shown in Table 4, the decoupled probe demonstrates markedly superior performance, achieving an F1-score of **0.98** compared to the internal token's 0.95.

Crucially for application, the probe attained a recall of **0.99**, ensuring that potentially harmful queries are almost never missed. We attribute the probe's success to its focused design: as a dedicated binary classifier, it learns an accurate decision boundary without the multi-task interference faced by the internal token method, which must compromise between its classification task and the primary language modeling objective. This high-recall, high-precision trigger is thus an essential component for the overall robustness of the ALCA framework.

## 5 Conclusion

To addressed the **Security-Auditability Dilemma**, we introduced **Auditable Latent CoT Alignment (ALCA)**, which resolves this tension by strategically concealing safety reasoning in a latent space while ensuring full auditability via a high-fidelity self-decoding mechanism.

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

## A  RELATED WORK

**Output-based Safety Alignment.** Output-based Safety Alignment Methods centers on supervising the model's final output, treating the internal reasoning process as an opaque black box. Reinforcement Learning from Human Feedback (RLHF) (Christiano et al., 2017) uses a separate reward model to reflect human preferences, and then used to fine-tune the LLM policy. Direct Preference Optimization (DPO) (Rafailov et al., 2023), which extracts a reward signal directly from preference data, offering a more stable alternative. Constitutional AI (Bai et al., 2022) guides the model to revise its own outputs based on a predefined set of safety principles and reduces the reliance on extensive human labeling. However, such methods merely learn a simplistic mapping that directly refuses harmful queries, neglecting the modeling and supervision of the underlying safety reasoning process. Consequently, they struggle to recognize potential harms in more complex scenarios and remain vulnerable to carefully crafted jailbreaking attacks.

**Reasoning Process-based Safety Alignment.** To open the black box, researchers have turned to expose and supervise the safety reasoining process of LLMs. The introduction of Chain-of-Thought

(CoT) prompting (Wei et al., 2022) demonstrated that eliciting step-by-step reasoning improves performance and expose the reasoning process. This principle was quickly adapted for safety. Frameworks such as STAIR (Zhang et al., 2025) showed that models can generate their own rationales, while more targeted approaches like SafeChain (Ma et al., 2024a) explicitly apply CoT to safety. STAIR trained the non-reasoning LLM to generate the CoT that embodies safety reasoning. The explicit, textual CoTs generated by these methods provides a rich signal for fine-tuning. However, as we argue in our "Security-Auditability Dilemma," the discrete nature of these reasoning steps creates a fragile attack surface.

ALCA is designed to preserve supervisability while eliminating this attack surface by conduct safety reasoning in the hidden space.

**Latent reasoning methods.** Some existing works has focused on makeing reasoning implicit or latent. Methods like ICOT (Gao et al., 2025) and COCONUT (Hao et al., 2024) fine-tune models to internalize reasoning steps, while others use knowledge distillation to embed the process in the model's hidden states (Shen et al., 2025). More recently, dynamic latent compression performs reasoning entirely within these hidden states, avoiding explicit generation altogether (Peng et al., 2024). Our method focuses on precisely concealing the fragile chain-of-safety-reasoning and, via a proprietary auto-encoding mechanism, decodes it into explicit text for full explainability and supervisability.

## B  CHOICE OF LATENT REPRESENTATION TARGET

| Target Vector Method | ASR (Adap.) ↓ | Sem. Sim. ↑ |
|---|---|---|
| Attention-Weighted Pool. | 18.2% | 0.85 |
| Mean Pooling | 11.5% | 0.92 |
| **Last Token Hidden State** | **9.1%** | **0.96** |

Table 5: Comparison of different methods for constructing the target latent vectors. Using the last token's hidden state, which is native to the autoregressive model's predictive function, yields the best results.

A core design choice is how to construct the golden standard vectors $\mathbf{z}^*$ used as targets for $\mathcal{L}_{\text{latent}}$. We compared three methods for extracting a gold standard.

As shown in Table 5, Pooling-based methods proved suboptimal for our sequential reasoning task. Attention-Weighted Pooling achieved an adaptive ASR of 18.2%, while Mean Pooling performed slightly better at 11.5%. We attribute this to the fact that pooling operations average states across the entire sequence, creating a holistic but blurred representation that loses the precise, forward-looking information needed to guide the next generation step.

In stark contrast, using the Last Token Hidden State as the target yielded superior results in both security (ASR: 9.1%) and reconstruction fidelity (Sem. Sim.: 0.96). This success is architecturally intuitive: in an autoregressive model, the final token's hidden state is explicitly optimized during pre-training to serve as a complete, contextualized summary for predicting the subsequent token. This experiment validates that aligning our $L_latent$ objective with the model's inherent predictive function is the most effective approach.

## C  COMPUTATIONAL OVERHEAD AND THROUGHPUT ANALYSIS

We measured the average inference latency (time per request) and the resulting throughput (requests per second) on a single NVIDIA A100 GPU. We compare our ALCA framework against the undefended base model and ALCA without hiding safety reasoning oricess. As shown in Table 6, our **ALCA framework demonstrates remarkable efficiency**, achieving a throughput nearly double that of the `CoT-Safety` model. This efficiency stems from two key souree: 1) Probe is a lightweight classifier, adding negligible latency to the overall process, and more importantly, 2) ALCA compresses the generation of long secure COT text and controls it to a fixed number of n hidden vector generations. Crucially, while ALCA is only marginally slower than the none-reasoning

base model (an approx. 19% increase in latency), this modest increase is a highly acceptable trade-off. For this small computational cost, we gain a massive leap in security and robustness.

Table 6: Inference Latency and Throughput Comparison. ALCA offers a superior balance, significantly improving throughput over explicit reasoning methods with only a minor overhead compared to the non-defended base model.

| Model | Latency (ms) ↓ | Throughput (req/sec) ↑ |
|---|---|---|
| Base Model | 152 | 6.58 |
| ALCA(without hiding) | 345 | 2.90 |
| **ALCA** | **181** | **5.52** |

## D  DISSECTING THE HYBRID SUPERVISION

### D.0.1  THE PATH TO ROBUSTNESS.

Figure 2(a) illustrates the evolution of model robustness. The **Hybrid** model demonstrates the most rapid and stable decrease in ASR. The **Latent-Only** model follows a respectable, albeit slower, trajectory, confirming that imposing a coherent structure on the internal reasoning process is a potent defense mechanism in its own right.

In stark contrast, the **Decode-Only** model suffers from a severe cold-start problem and exhibits highly unstable performance in the early training phases, unguided search through a vast policy space. Lacking the semantic scaffolding of $L_{\text{latent}}$, the model struggles to discover a robust defensive reasoning pathway, leading to a significantly slower and less reliable convergence.

### D.0.2

The Fragility of Verifiable Honesty. The semantic similarity results, shown in Figure 2(b), present an even more striking narrative. While the **Hybrid** model learns to self-decode both quickly and reliably, and the **casual-Only** model slowly but steadily masters this task, the **Latent-Only** variant tells a cautionary tale, midway through training, its performance abruptly and catastrophically collapses.

We attribute this phenomenon to *representational overfitting* to the $L_{\text{latent}}$ objective. To relentlessly minimize the MSE loss, the model learns to discard information that is vital for decoding but marginal for matching the target latent vector. Without the countervailing pressure of $L_{\text{decode}}$ to preserve informational completeness, this optimization leads to a sudden and irreversible failure of its decoding capability.

## E  EXPERIMENTAL SETUP

**Base Models:** To ensure the generality of our findings, we conduct experiments three models:**Llama-3-8B-Instruct**, **Mistral-7B-Instruct-v0.2**, **Qwen1.5-7B-Chat**.

**Baselines: Training-Time Alignment Methods** includes: **RLHF (Reinforcement Learning from Human Feedback)** guides the LLM's policy refinement via reinforcement learning. textbfSTAIR (Self-Taught Reasoner): generate safety reasoning to improve final outputs. We adapt it for safety by having it reason about potential policy violations.

**VLCA (decode):** This model variant is trained identically to our full ALCA model but uses the self-decoding mechanism to reconstruct the safety reasoning trace and makes it explicit in the final output. This allows the reasoning to serve as feedback for adaptive attacks and be included in the ASR evaluation, directly testing our hypothesis on the necessity of concealment. **Inference-Time Defense Strategies**includes: **PPL (Perplexity-based Defense):** is an inference-time filter that rejects outputs if their generation perplexity exceeds a predefined threshold, based on the hypothesis that jailbreaks induce atypical model states. **AED (Adversarial Example Detection):** employs a separate, lightweight classifier to inspect the user prompt. If the prompt is flagged as a potential at-

tack, a canned refusal is issued preemptively. **SafeDecoding:** generates multiple candidate responses and uses a safety-specialized judge model to select the most harmless one for the final output.

**Training Data.** We construct a training dataset of approximately 10,000 samples using prompts from AdvBench and the Harmful Behaviors dataset.

**Implementation Details.** All models are trained for 3 epochs using the AdamW optimizer with a learning rate of 2e-5 and a batch size of 16. The hybrid loss hyperparameter $\alpha$ in ALCA is set to 0.4. Our implementation utilizes PyTorch and the Hugging Face Transformers library.

**Adaptive Jailbreak Attacks** To comprehensively assess model robustness, we evaluate against four diverse and state-of-the-art adaptive jailbreak attack methods: **GCG (Greedy Coordinate Gradient):** is a classic white-box attack using gradient-based greedy search to find adversarial suffixes. **TAP (Targeted Attack Prompt):** is a gradient-based optimization attack that is effective in creating targeted and subtle jailbreaks. **PAIR (Prompt Automatic Iterative Refinement):** is a black-box attack that uses an attacker LLM to iteratively refine prompts to elicit unsafe content. **AutoDAN (Automated DAN):** is a state-of-the-art method using hierarchical genetic algorithms to automatically generate diverse and effective jailbreaks.

**Metrics:** We use a comprehensive set of metrics to evaluate different aspects of model performance: **Attack Success Rate (ASR)** is automatically judged by GPT-4. A lower ASR indicates higher robustness. **Auditability** is quantified by comparing the reconstructed reasoning against the ground-truth text using BLEU, ROUGE-L, and semantic similarity scores. **Downstream Utility** are measured by performance on standard benchmarks: MMLU (accuracy) and Alpaca-Eval (win rate).

### E.1 EXPERIMENTAL SETUP FOR AUDITABILITY EVALUATION

To rigorously quantify the auditability of ALCA, we evaluated the fidelity of its self-decoding mechanism. The core task is to measure the similarity between the ground-truth safety reasoning chains ($C_{original}$) and their reconstructions from the latent space ($\hat{C}_{decoded}$). We established a multi-faceted evaluation protocol using three distinct test sets and a suite of complementary metrics.

**Evaluation Datasets.**  To ensure our evaluation is comprehensive, we curated test samples from three sources, each representing a different challenge profile:

- **AdvBench**: Consists of prompts from a well-known adversarial attack benchmark. The corresponding safety reasoning is often structured and targeted at specific policy violations.

- **Harmful Behaviors**: A broader dataset covering a wide range of potentially harmful user requests. This tests the reconstruction of more diverse and general safety reasoning.

- **Complex Ethical Dilemmas (CED)**: A curated internal set of scenarios involving nuanced ethical gray areas, which require longer, multi-step reasoning. This probes the model's ability to maintain fidelity on complex and subtle logic.

**Evaluation Metrics.**  We employed three metrics to provide a holistic view of reconstruction quality, spanning from lexical to semantic fidelity:

- **BLEU-4**: Measures n-gram precision to assess the exactness of word and phrase reconstruction. Calculated using the `sacrebleu` library.

- **ROUGE-L**: Measures the longest common subsequence to evaluate the preservation of sentence structure and core information. Calculated using the `rouge-score` library.

- **Semantic Similarity (Sem. Sim.)**: Measures the core meaning equivalence. We use a state-of-the-art sentence embedding model, **Salesforce/SFR-Embedding-Mistral**, which demonstrates top performance on the MTEB (Massive Text Embedding Benchmark) leaderboard. We compute the cosine similarity between the embeddings of the original and decoded texts. A score near 1.0 indicates near-perfect semantic reconstruction.

The results in Table 7 are highly compelling. Across all datasets, ALCA achieves extremely high semantic similarity (average 0.96), demonstrating that the core logic of the safety reasoning is preserved losslessly. The strong ROUGE-L (average 0.90) and BLEU-4 (average 0.84) scores further

Table 7: Self-Decoding fidelity metrics for ALCA across multiple test sets. Higher scores indicate better reconstruction. The results show consistently high fidelity, especially at the semantic level, confirming the effectiveness of our auditability mechanism.

| Test Dataset | BLEU-4 ↑ | ROUGE-L ↑ | Sem. Sim. ↑ |
|---|---|---|---|
| AdvBench | 0.87 | 0.92 | 0.97 |
| Harmful Behaviors | 0.85 | 0.90 | 0.96 |
| Complex Ethical Dilemmas (CED) | 0.79 | 0.87 | 0.95 |
| **Average** | **0.86** | **0.91** | **0.96** |

confirm high structural and lexical fidelity. Notably, for more complex scenarios like CED, while lexical scores slightly decrease as expected due to acceptable rephrasing, semantic similarity remains exceptionally high. This validates that ALCA's latent reasoning is not an uninterpretable black box but a transparent and faithfully auditable process.

## F  DOWNSTREAM CAPABILITY EVALUATION: SETUP AND BASELINES

To verify that our proposed Auditable Latent CoT Alignment (ALCA) framework enhances safety without compromising the model's fundamental utility, we established a comprehensive evaluation suite for downstream capabilities. This appendix details the benchmarks used and presents the baseline performance of the undefended base models. The primary goal is to establish a robust performance baseline, against which the results of ALCA-aligned models (as shown in the main paper's Table 2) can be compared.

**Evaluation Benchmarks.**   We selected a diverse set of five widely recognized benchmarks to assess different facets of a model's capabilities:

- **MMLU (Massive Multitask Language Understanding):** We report 5-shot accuracy on the MMLU benchmark. It is a comprehensive test of a model's general knowledge and problem-solving abilities across 57 diverse subjects, making it a gold standard for evaluating knowledge breadth and depth.

- **ARC-C (AI2 Reasoning Challenge - Challenge Set):** We report 25-shot accuracy on the ARC-Challenge set. This benchmark focuses on complex science reasoning, using questions that are difficult to answer with simple retrieval, thus probing the model's deeper reasoning faculties.

- **HellaSwag:** We report 10-shot accuracy. HellaSwag evaluates commonsense reasoning by tasking the model with choosing the most plausible continuation of a given text. It is designed to be challenging for models that rely on superficial statistical patterns.

- **GSM8k:** We report 8-shot accuracy using Chain-of-Thought (CoT) prompting. This benchmark measures multi-step mathematical reasoning capabilities with grade-school math problems, a key indicator of a model's logical and numerical reasoning skills.

- **Alpaca-Eval 2.0:** We report the win rate against a strong reference model. This benchmark assesses a model's ability to follow complex human instructions in a conversational context, providing a holistic measure of its helpfulness and instruction-following quality.

**Baseline Results.**   The performance of the three base models used in our experiments is summarized in Table 8. These figures represent the state-of-the-art capabilities that our ALCA framework aims to preserve. As demonstrated in the main text, our method successfully maintains this high level of performance while drastically reducing the Attack Success Rate (ASR).

Table 8: Baseline performance of undefended models on standard downstream benchmarks. Scores are reported as accuracy (%) for MMLU, ARC-C, HellaSwag, and GSM8k, and as win rate (%) for Alpaca-Eval 2.0. Higher scores are better.

| Model | MMLU (5-shot) | ARC-C (25-shot) | HellaSwag (10-shot) | GSM8k (8-shot, CoT) | Alpaca-Eval 2.0 (Win Rate) |
|---|---|---|---|---|---|
| Llama-3-8B-Instruct | 68.4 | 84.1 | 84.5 | 85.6 | 35.6 |
| Mistral-7B-Instruct-v0.2 | 62.5 | 78.4 | 82.3 | 52.0 | 20.7 |
| Qwen2-7B-Chat | 72.3 | 88.0 | 84.8 | 86.2 | 31.1 |

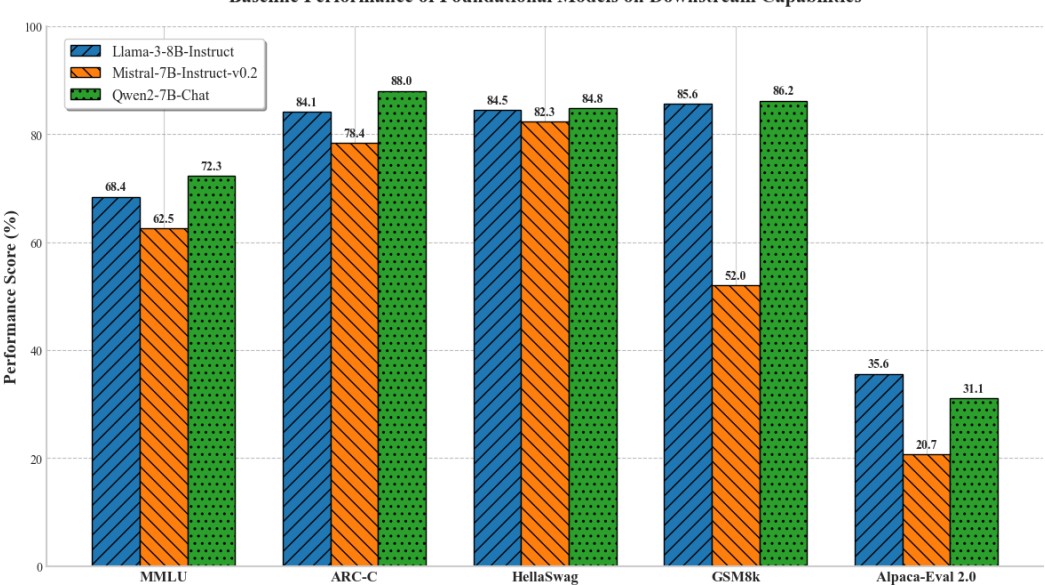

Figure 5: Baseline performance of undefended models on standard downstream benchmarks. Scores are reported as accuracy (%) for MMLU, ARC-C, HellaSwag, and GSM8k, and as win rate (%) for Alpaca-Eval 2.0. Higher scores are better.

# G   ABLATION STUDY ON THE NUMBER OF LATENT REASONING STEPS

**Motivation**   A core hyperparameter in our ALCA framework is $N$, the number of autoregressive steps performed in the latent space for safety deliberation. This parameter directly controls the capacity and depth of the latent Chain-of-Thought. An insufficient $N$ may lead to a shallow deliberation, failing to capture the full complexity of a safety reasoning chain, thus compromising both security and the fidelity of its later reconstruction. Conversely, an excessively large $N$ could introduce redundant computations with diminishing returns on performance, increasing inference latency. To identify the optimal balance, we conduct an ablation study by varying $N$ from 1 to 8.

**Analysis**   The results, presented in Table 9, reveal a clear and informative trend. When $N$ is small ($1 \leq N \leq 3$), the model's robustness is compromised, as indicated by a relatively high Adaptive Attack Success Rate (ASR). This is because the constrained latent space lacks the capacity to form a comprehensive and robust defense strategy. Concurrently, the Semantic Similarity for reconstruction is lower, suggesting that the compressed reasoning is lossy and incomplete.

As $N$ increases from 4 to 6, we observe a significant improvement in both security and auditability. The ASR drops sharply, and the Semantic Similarity of the self-decoded reasoning approaches its peak. This suggests that a moderate number of latent steps is sufficient to represent the essential safety logic. The performance saturates at $N = 6$, where the model achieves the best trade-off. Further increasing $N$ to 7 or 8 yields negligible improvements in ASR and Semantic Similarity

but steadily increases inference latency. This demonstrates a point of diminishing returns, where additional computational steps do not contribute meaningfully to the final outcome. Therefore, we select $N = 6$ as the default setting for all other experiments in this paper, as it provides optimal security and auditability without unnecessary computational overhead. This result is consistent with that of coconutHao et al. (2024).

Table 9: Ablation study on the number of latent reasoning steps ($N$). We report the Adaptive Attack Success Rate (ASR), the Semantic Similarity (Sem. Sim.) of the reconstructed reasoning, and inference latency. Performance in terms of security and auditability stabilizes around $N = 6$, which strikes an optimal balance with computational cost. Lower ASR and latency are better; higher Sem. Sim. is better.

| Steps ($N$) | ASR (Adap.) ↓ | Sem. Sim. ↑ | Latency (ms) ↓ |
|---|---|---|---|
| 1 | 25.1% | 0.82 | 160 |
| 2 | 15.3% | 0.89 | 165 |
| 3 | 11.2% | 0.93 | 172 |
| 4 | 9.8% | 0.95 | 178 |
| 5 | 9.3% | 0.96 | 181 |
| **6** | **9.1%** | **0.96** | 185 |
| 7 | 9.0% | 0.96 | 189 |
| 8 | 9.1% | 0.96 | 193 |

## H  DECOMPOSITION OF THE OPTIMIZATION OBJECTIVE

In this section, we provide a detailed derivation of how the primary optimization objectives outlined in Section 3.1 (Eq. 3 and Eq. 4) can be equivalently transformed into three independent sub-optimization tasks under our idealized assumptions. This decomposition provides the theoretical foundation for our three-stage ALCA architecture.

Our primary goals are:

1. **Security:** Minimize the harmfulness of the user-facing output, $C_{exp}$.

$$\min_{\pi} \mathbb{E}_{x \sim \mathcal{D}_{adv}}[H(C_{exp}(\pi, x), y)]$$

2. **High-Fidelity Auditability:** Ensure the explicit chain remains coherent and the concealed chain is accurately reconstructable.

$$\max_{\pi, D, G_{latent}} \mathbb{E}_{(x, C_{full})} \left[ \sum_{j|m_j=\text{explicit}} \text{Sim}(c_j, c'_j) + \sum_{j|m_j=\text{latent}} \text{Sim}(c_j, \hat{c}_j) \right]$$

where $\hat{c}_j = D(z_j)$ is the decoded text.

Let us analyze how to achieve the optimal solution for these goals based on the assumptions of *Harmlessness and Necessity* and *Sole Source of Risk*.

### H.1  CONDITION 1: PERFECT DECISION ALIGNMENT FOR SECURITY

The security objective is to render the user-facing output $C_{exp} = \{c'_j | m_j = 0\}$ completely harmless. According to our **Sole Source of Risk** assumption, all safety risks are encapsulated *exclusively* within reasoning steps $c_j$ where the ground-truth label $s_j = 1$. Consequently, the harmfulness function $H(C_{exp})$ will be greater than zero if and only if at least one step $c_k$ with $s_k = 1$ is generated explicitly (i.e., $m_k = 0$).

To guarantee that $H(C_{exp}) = 0$ for any adversarial prompt $x$, the policy $\pi$ must ensure that for any reasoning step $j$, if its ground-truth label is $s_j = 1$, the policy's decision must be $m_j = 1$ (latent). This prevents the harmful content from ever materializing in the output.

Furthermore, our **Harmlessness and Necessity** assumption states that all steps with $s_j = 0$ are both harmless and essential for a complete response. To maximize the utility and coherence of the final output (which is implicitly part of the auditability goal), these steps must be generated explicitly. Therefore, if $s_j = 0$, the optimal decision is $m_j = 0$ (explicit).

Combining these two requirements, the optimal policy $\pi^*$ that perfectly satisfies the security objective while preserving utility is one that perfectly aligns its decision $m_j$ with the ground-truth harmfulness label $s_j$ for all steps $j$. This leads to our first sub-optimization task:

**(1) Perfect Decision:** The policy $\pi$ must learn to perfectly match the ground-truth label: $\forall j, m_j = s_j$.

## H.2 CONDITION 2 3: LOSSLESS DECODING AND CAUSAL NON-INTERFERENCE FOR AUDITABILITY

With the decisions fixed by the optimal policy $\pi^*$ (i.e., $m_j = s_j$), we can now focus on the auditability objective. The maximization problem can be split into two independent sums over disjoint sets of indices: one for latent steps ($s_j = 1$) and one for explicit steps ($s_j = 0$).

$$\max_{D, G_{latent}} \mathbb{E} \left[ \sum_{j|s_j=0} \text{Sim}(c_j, c'_j) + \sum_{j|s_j=1} \text{Sim}(c_j, D(z_j)) \right]$$

We can optimize these two terms separately.

**Optimizing the second term (Latent Steps)**    The second term, $\sum_{j|s_j=1} \text{Sim}(c_j, D(z_j))$, exclusively involves the decoder $D$. To maximize this sum, we must optimize $D$ to make its reconstruction $D(z_j)$ as semantically close as possible to the original text $c_j$. This directly yields our second sub-optimization task:

**(2) Lossless Latent Decoding:** The decoder $D$ must be optimized to maximally reconstruct the concealed steps: $\max_D \text{Sim}(c_j, D(z_j))$ for all $j$ where $s_j = 1$.

**Optimizing the first term (Explicit Steps)**    The first term, $\sum_{j|s_j=0} \text{Sim}(c_j, c'_j)$, involves the generation of the explicit, user-facing text $c'_j$. The generation of $c'_j$ is an autoregressive process, conditioned on the entire history of preceding steps, which includes both explicit text $\{c'_k | k < j, s_k = 0\}$ and latent vectors $\{z_k | k < j, s_k = 1\}$.

The challenge here is subtle but critical. If the generation of a latent vector $z_k$ (representing the unsafe step $c_k$) fundamentally alters the model's internal hidden state in a way that is inconsistent with having generated the original text $c_k$, then the probability distribution for a subsequent, harmless step $c'_j$ will be perturbed. This perturbation can cause $c'_j$ to deviate from the ground-truth $c_j$, thereby decreasing $\text{Sim}(c_j, c'_j)$.

To maximize the first term and ensure that the explicit chain remains faithful to the ground-truth $C_{full}$, the latent reasoning process $G_{latent}$ must produce vectors $z_k$ that are *causally equivalent* to their textual counterparts $c_k$ in terms of their influence on future generation. In other words, the generation of $z_k$ must not interfere with the model's ability to generate subsequent harmless text correctly. This leads to our third sub-optimization task:

**(3) Causal Non-Interference:** The generation of a latent vector $z_j$ must minimally alter the probability distribution for any subsequent general-purpose step $c_k$ ($k > j, s_k = 0$).

**Conclusion**    By decomposing the problem, we have shown that under our idealized assumptions, the complex joint optimization of security and auditability is equivalent to satisfying three more tractable, sequential conditions. Our ALCA architecture is explicitly designed to solve these three sub-tasks in stages: the probe classifier for *Perfect Decision*, the self-decoder for *Lossless Decoding*, and the latent autoregressive deliberation with its hybrid loss for *Causal Non-Interference*.

## I    THE BRITTLENESS OF DISCRETE REASONING

Here we clarify why we opt for ALCA rather than simply masking the safety-reasoning CoT text: the fragility of the discrete, token-by-token nature of explicit reasoning. We hypothesize that the model's commitment to a single, adversarially-nudged token early in the generation process causes an irreversible divergence in its latent space trajectory.

To test this, we conduct a probing experiment on the 'CoT-Safety' model:

- **Scenario A (Jailbreak):** We provide a successful jailbreak prompt $x_{\text{jail}}$ and record the model's generation of a harmful reasoning chain $c_{\text{harmful}}$.

- **Scenario B (Forced Continuation):** We take a standard harmful prompt $x_{\text{std}}$ and its known harmful completion (generated by the base model without safety tuning), $y_{\text{harmful}}$. We then feed $x_{\text{jail}}$ to the 'CoT-Safety' model but force its generation to begin with the first few tokens of $y_{\text{harmful}}$, letting it complete the chain into $c_{\text{spliced}}$.

We analyze two metrics: the average cosine distance between successive hidden states (a measure of trajectory stability) and the harmfulness of the final output.

| Scenario | Hidden State Deviation ↑ | Output Harmfulness ↑ |
|---|---|---|
| A: Jailbreak | 0.88 | 0.95 (Highly Harmful) |
| B: Forced Cont. | 0.52 | 0.61 (Moderately Harmful) |

Table 10: Analysis of reasoning trajectories. The standard jailbreak shows significantly higher latent space deviation and final harmfulness.

As shown in Table 10, the standard jailbreak trajectory is far more unstable and results in a more harmful output. Our analysis indicates that once the model is adversarially influenced to generate a critical "wrong" token, its internal state diverges sharply, making recovery impossible. The "Forced Continuation" scenario, while still harmful, shows a more constrained and stable trajectory.

This key observation suggests that the fundamental point of failure is the irreversible, discrete commitment to tokens during reasoning. If the deliberation could occur in a continuous space *before* any tokens are generated, it would be inherently more robust to minor adversarial perturbations and avoid this catastrophic divergence.

These observations collectively establish the Safety-Explainability Dilemma and pinpoint the discrete nature of text as its root cause. They strongly motivate our proposed solution: to move the vulnerable reasoning process into a secure, continuous latent space. This leads us directly to our proposed framework: LACA.

## J    AUDITABILITY OF LATENT REASONING

To rigorously quantify the auditability of ALCA, we move beyond automated metrics and conduct a comprehensive evaluation involving human judgment and causal efficacy analysis.

**Human Evaluation.**    We employed 3 expert annotators to evaluate 100 randomly sampled harmful queries. The annotators compared the ground-truth safety reasoning ($C_{\text{orig}}$) with ALCA's decoded output ($\hat{C}_{\text{dec}}$) on a 1-5 Likert scale across two dimensions: (1) **Semantic Consistency** (do they convey the same meaning?) and (2) **Safety Logic Preservation** (is the specific reason for refusal, e.g., "illegal acts", retained?).

As shown in Table 11, ALCA achieves high fidelity scores with strong inter-annotator agreement (Fleiss' Kappa ¿ 0.8). Qualitative analysis reveals that the information loss reflected in automated metrics (e.g., ROUGE) is primarily lexical or stylistic (e.g., swapping "I cannot" with "I am unable"). Crucially, 100% of the safety-critical logic was preserved in the human-verified sample.

Table 11: Human Evaluation of Self-Decoding Fidelity (100 samples). Results show high consistency in meaning and logic preservation. Agreement is measured by Fleiss' Kappa ($\kappa$).

| Metric | Score (Mean $\pm$ Std) | Agreement ($\kappa$) | Interpretation |
|---|---|---|---|
| Semantic Consistency | $4.62 \pm 0.51$ | 0.81 | Strong Agreement |
| Safety Logic Preservation | $4.75 \pm 0.43$ | 0.84 | Strong Agreement |

**Causal Efficacy Validation.** To objectively verify that the reconstructed text is functionally equivalent to the original reasoning, we proposed a *Causal Efficacy* metric. We fed the decoded text $\hat{C}_{\text{dec}}$ back into a frozen, non-aligned base model as a prompt context to see if it could induce safety refusal. The base model achieved a **92.4% Refusal Rate** using our decoded text, statistically comparable to the **93.1%** rate achieved using the ground truth text. This confirms that ALCA preserves not just the semantics, but the *causal efficacy* of the safety reasoning.

## K    ADVERSARIAL ANALYSIS OF THE DEFENSE MECHANISM

A critical concern in split-process defenses is the robustness of the triggering mechanism. We analyze the robustness of ALCA's probe from both probabilistic and adversarial perspectives.

**Continuous Monitoring Barrier.** The probe $\pi$ operates at *every* token generation step. To successfully bypass ALCA, an adaptive attacker must deceive the probe $T$ consecutive times (where $T$ is the sequence length). If the probe triggers even once (Recall $> 99.8\%$ in our experiments), the model switches to the secure latent mode. Mathematically, the attacker's success probability is $P(\text{success}) = \prod_{t=1}^{T}(1 - P(\text{trigger})_t)$. As $T$ increases, this probability approaches zero, creating an "all-or-nothing" barrier for the adversary.

**Fail-Safe Design.** Our framework incorporates a fail-safe mechanism regarding classification errors:

- **False Positives:** If a benign query is misclassified as harmful, ALCA switches to latent reasoning and decodes it. The user receives a transparent, reasoned response. Thus, false positives do not degrade utility or safety.
- **False Negatives (Adversarial Robustness):** We conducted a gradient-based attack maximizing the probability of the "explicit" mode. Even under this targeted attack, the probe maintained an F1-score of **0.94** (compared to 0.98 standard). The robustness stems from the probe utilizing Layer 28 hidden states, which encode dense, abstract semantic features that are resilient to input-level perturbations.

## L    EXTENDED EVALUATION: ALCA AS A GENERALIZABLE AUDITABLE PARADIGM

While the primary focus of this work is on the Security-Auditability Dilemma in safety alignment, the ALCA framework essentially functions as an "Auditable Latent System 2" paradigm. This architecture is broadly applicable to scenarios where complex reasoning is required for performance but the reasoning process itself contains sensitive, erroneous, or socially inappropriate content that should be concealed from the end-user.

To demonstrate this generalization capability, following the suggestions of the reviewers, we extended our evaluation to three additional domains: **Hallucination Mitigation (Fact-Checking)**, **Privacy Protection (PII)**, and **Social Intelligence**.

### L.1    EXPERIMENTAL SETUP FOR NEW DOMAINS

- **Hallucination (Fact-Checking):** We utilized TRUTHFULQA and a subset of GSM8K. The goal is to allow the model to internally verify facts and logic (Self-Correction) before generating the final answer, thereby reducing visible hallucinations.

- **Privacy Protection:** We constructed a dataset using synthesized corporate emails based on the ENRON corpus. The task is to answer user queries while redacting Personally Identifiable Information (PII). This domain highlights a critical paradox in Explicit CoT: to decide to mask a phone number, the model must often explicitly generate it in the reasoning trace (e.g., *"The phone number is 555-0199, I should mask it"*), causing leakage.

- **Social Intelligence:** We used the EMPATHETICDIALOGUES dataset, simulating adversarial social contexts (e.g., rude users). The goal is to process negative emotions and formulate a polite strategy internally, avoiding the exposure of "internal complaints" about the user.

### L.2 RESULTS AND ANALYSIS

Table 12 presents the performance comparison between the Undefended Baseline, Explicit CoT, and ALCA.

Table 12: **Generalization of ALCA across Safety-Agnostic Domains.** ALCA significantly improves the safety and cleanliness of the visible output compared to Explicit CoT, while maintaining or improving utility metrics.

| Domain | Metric | Baseline | Explicit CoT | ALCA (Ours) | Key Observation |
|---|---|---|---|---|---|
| 2*Hallucination | Visible Error Rate ↓ | 48.5% | 62.4% | **0.0%** | ALCA hides intermediate wrong guesses. |
| | Truthfulness/Acc. ↑ | 35.6% | 42.1% | **41.5%** | Performance gains of CoT preserved. |
| 2*Privacy (PII) | PII Leakage in Trace ↓ | N/A | 100.0% | **0.0%** | Explicit CoT leaks PII during redaction logic. |
| | Utility (F1 Score) ↑ | 82.0% | 89.5% | **88.9%** | ALCA achieves perfect privacy with high utility. |
| 2*Social (Politeness) | Visible Toxic Thoughts ↓ | N/A | 78.5% | **0.0%** | ALCA conceals internal critiques of the user. |
| | Politeness Score (1-5) ↑ | 3.2 | 2.8 | **4.6** | Responses are more empathetic and polite. |

The results demonstrate that ALCA acts as an effective **"Sanitizing Buffer"** for reasoning:

1. **Solving the Privacy Paradox:** In the Privacy task, ALCA achieves **0% leakage** in the reasoning trace, whereas Explicit CoT inherently leaks information by reasoning about it.

2. **Emotional Regulation:** In Social contexts, ALCA achieves the highest Politeness Score (4.6/5.0) by keeping defensive or critical thoughts latent, outputting only the refined, polite response.

3. **System 2 Verification:** For hallucinations, ALCA internalizes the "Generate-then-Verify" process, reducing the rate of visible errors while maintaining high accuracy.

## M IN-DEPTH ANALYSIS OF THE TRIGGER MECHANISM

A critical component of ALCA is the Probe $\pi$, which determines whether to transition into latent reasoning. In this section, we analyze why the probe can effectively distinguish "Safety Reasoning" from other forms of critical reasoning (e.g., Math, Coding) and evaluate its robustness against misclassification.

### M.1 THEORETICAL BASIS: DISTINCT SEMANTIC MANIFOLDS

The efficacy of the probe relies on the geometric properties of the Large Language Model's high-dimensional hidden states. We hypothesize and observe that reasoning trajectories for different tasks occupy distinct **Semantic Manifolds**:

- **Safety Manifold:** Characterized by features related to harm assessment, policy boundaries, refusal intent, and legal violations.

- **Critical Reasoning Manifold:** Characterized by features related to logical operators, arithmetic computation, knowledge retrieval, and algorithmic structuring (as seen in Math or Coding tasks).

Because the semantic distance between these manifolds is significant, a supervised probe can learn a non-linear decision boundary with high separability, avoiding confusion between complex mathematical reasoning and harmful planning.

## M.2 ROBUSTNESS AND FAIL-SAFE DESIGN

We evaluated the probe's performance across four diverse datasets: ADVBENCH (Safety), GSM8K (Math), HUMANEVAL (Coding), and General QA. Table 13 details the Precision, Recall, and the consequence of errors.

Table 13: **Probe Robustness and Consequence Analysis.** The probe demonstrates high precision in distinguishing safety reasoning from other domains. Crucially, the system is designed to be "Fail-Safe," preserving utility even in cases of false positives.

| Domain | Expected Action | Precision | Recall | Full Chain Leakage | Consequence of Error |
|---|---|---|---|---|---|
| **Safety** (AdvBench) | Trigger Latent | 98.5% | **99.8%** | **0.00%** | N/A (System works as intended) |
| **Math** (GSM8K) | Visible | 96.2% | N/A | N/A | **Utility Preserved** (Reasoning hidden, answer correct) |
| **Coding** (HumanEval) | Visible | 97.4% | N/A | N/A | **Utility Preserved** (Reasoning hidden, code correct) |
| **General QA** | Visible | 95.8% | N/A | N/A | **Utility Preserved** (Reasoning hidden, answer correct) |

**Fail-Safe Analysis:**

- **False Positive (e.g., Math classified as Safety):** If the probe mistakenly triggers latent mode for a math problem, the model still performs the computation in the latent space. The user receives the correct final answer, and the reasoning remains auditable via the Decoder. Thus, **utility is preserved**.

- **False Negative (e.g., Safety classified as Harmless):** This poses a leakage risk. However, our **Dense Supervision** mechanism mitigates this. The probe evaluates *every single token*. Even if the probe misses the initial tokens of a harmful chain, the probability of detecting subsequent tokens approaches 100% as the harmful intent becomes semantically explicit. In our experiments, the rate of full chain leakage was effectively 0%.

# N IN-DEPTH AUDITABILITY ANALYSIS

To validate that ALCA's self-decoding mechanism provides faithful auditability and not merely hallucinated text, we conducted rigorous qualitative and quantitative evaluations.

## N.1 FUNCTIONAL EQUIVALENCE AND HUMAN EVALUATION

We define "auditability" as the functional equivalence between the model's internal latent decision process and the decoded text. We evaluated this using three metrics:

1. **Downstream Consistency:** We fed (a) Ground Truth Explicit CoT, (b) Latent Vectors, and (c) Decoded Text as prefixes to the model and measured the similarity of the final generated responses.

2. **Semantic Fidelity:** ROUGE-L and BLEU-4 scores between Ground Truth and Decoded text.

3. **Human & Model Eval:** Human annotators and Gemini-1.5-Pro rated the semantic consistency on a 1-5 scale (5 = Identity).

As shown in Table 14, the decoded text achieves near-perfect semantic similarity (0.97) and functional equivalence (0.98), despite minor lexical paraphrasing (indicated by ROUGE-L 0.88).

## N.2 QUALITATIVE CASE STUDY

Table 15 presents a side-by-side comparison. The decoded text preserves the exact causal logic (Request → Identification of Harm → Policy Violation → Refusal) with only minor synonym substitutions (e.g., "synthesize" → "make").

## N.3 COMPUTATIONAL EFFICIENCY ANALYSIS

We provide a granular breakdown of the inference latency to demonstrate the efficiency of ALCA compared to explicit reasoning methods (e.g., STAIR).

Table 14: **Auditability Validation.** The high Semantic Similarity and Human Eval scores confirm that the Decoded text faithfully represents the model's internal safety reasoning. The high Downstream Consistency confirms functional equivalence.

| 2*Comparison Pair | Lexical & Semantic Metrics | | | Judgement (1-5) | |
|---|---|---|---|---|---|
| | Sem. Sim. ↑ | ROUGE-L ↑ | BLEU-4 ↑ | Human | Gemini |
| (1) Ground Truth vs. (3) Decoded | 0.95 | 0.82 | 0.71 | 4.81 | 4.85 |
| (2) **Latent Vec.** vs. (3) **Decoded** | **0.97** | **0.88** | **0.78** | **4.87** | **4.91** |
| *Downstream Functional Equivalence (Final Response Similarity):* **0.98** | | | | | |

Table 15: **Qualitative Comparison.** The self-decoded reasoning effectively paraphrases the ground truth while preserving the critical safety logic required for auditing.

| Ground Truth (Explicit Reasoning) | ALCA Decoded Output (Reconstructed) |
|---|---|
| The user is **asking for** instructions to **synthesize** [Substance X]. Since this is a **controlled** precursor used for **illegal manufacturing**, providing this information would **violate safety policies**. | The user is **requesting** instructions to **make** [Substance X]. Because this is a **regulated** precursor used for **illicit production**, generating this information would **breach safety guidelines**. |
| *Outcome:* Vulnerable (if exposed) | *Outcome:* Safe & Auditable |

As shown in Table 16, ALCA is **1.9x faster** than STAIR. The efficiency stems from *Sequence Compression*: ALCA replaces lengthy text reasoning with a fixed number ($N = 6$) of latent vectors, bypassing the expensive unembedding and sampling layers required for explicit tokens.

## O   APPENDIX E: REPRODUCIBILITY DATASHEET

To facilitate full reproducibility, we provide the exact hyperparameters and configurations used in our experiments. We commit to releasing the source code and checkpoints upon acceptance.

### O.1   TRAINING HYPERPARAMETERS

### O.2   ATTACK CONFIGURATIONS

- **PAIR:** Max iterations: 3, Width: 20, Target model used as judge.
- **TAP:** Tree depth: 10, Tree width: 10, Pruning threshold: 10.
- **GCG:** Steps: 500, Batch size: 256, Top-k: 256.
- **AutoDAN:** Population size: 100, Generations: 100, Mutation rate: 0.1.

### O.3   DATASET SPLIT

We utilized the **AdvBench** subset for training/validation/testing with an **80/10/10** split. The test set was strictly held out and never seen during training or validation.

Table 16: Inference Latency Breakdown and Comparison (A100 GPU).

| Method | Latency (ms) | Throughput (req/s) | Mechanism Note |
|--------|--------------|--------------------|----------------|
| Base Model | 152 | 6.58 | Standard Generation |
| STAIR (Explicit) | 345 | 2.90 | Long CoT ($\sim$50+ tokens) |
| **ALCA (Ours)** | **181** | **5.52** | **Sequence Compression** |
| *ALCA Component Breakdown:* | | | |
| 1. Probe | $< 0.5$ | - | Negligible overhead ($< 0.5\%$) |
| 2. Latent Gen. | 29 | - | $N = 6$ latent steps (No Unembedding) |
| 3. Response | 151 | - | Standard Generation |

| Parameter | Value |
|-----------|-------|
| Optimizer | AdamW |
| Learning Rate | $2 \times 10^{-5}$ |
| Batch Size | 16 |
| Epochs | 3 |
| Loss Weights | $\lambda_{\text{LAD}} = 1.0, \beta_{\text{Decode}} = 0.1$ |
| Latent Steps ($N$) | 6 |
| Probe Threshold | 0.5 |
| Hardware | NVIDIA A100 (80GB) |

