# OpenReview forum: "Resolving the Security-Auditability Dilemma with Auditable Latent Chain-of-Thought"
_ICLR.cc/2026/Conference — ICLR 2026 Conference Desk Rejected Submission_

### Official Review · Reviewer_d8ZB · 2025-10-24

**Soundness:** 2
**Presentation:** 1
**Contribution:** 2
**Rating:** 2
**Confidence:** 4

**Summary:**

This paper proposes **Auditable Latent Chain-of-Thought Alignment (ALCA)**, a framework that aims to balance *security* and *auditability* in safety-aligned large language models.
The authors argue that exposing explicit reasoning traces (Chain-of-Thoughts) improves transparency but simultaneously enables jailbreaks and prompt-injection attacks.
ALCA attempts to solve this by encoding reasoning in a **latent space**, inaccessible to adversaries, while providing an **auditing mechanism** that can decode latent representations into interpretable rationales for safety verification.
Experiments are conducted on several LLMs (LLaMA-3-8B, Mistral-7B, Qwen2-7B) using multiple jailbreak benchmarks (GCG, AutoDAN, PAIR).

**Strengths:**

- The paper raises an important and underexplored issue in model alignment: the inherent tradeoff between **security** and **auditability**.
- The idea of performing safety reasoning in a **latent space** is conceptually appealing and may inspire future research.
- Evaluation includes several modern jailbreak methods (GCG, AutoDAN, PAIR), which shows awareness of the current security landscape.
- The ablation experiments (latent-only vs. causal-only vs. hybrid) offer some insight into how different supervision components contribute to robustness.

**Weaknesses:**

- **Lack of experimental rigor:** Attack success rate results are not averaged across runs or accompanied by standard deviations.
- **Unclear evaluation metrics:** GPT-4-based judgments are used without assessing consistency or inter-run variability.
- **Incremental novelty:** The method builds upon prior latent reasoning and safe decoding techniques without introducing a fundamentally new idea.
- **Missing cost analysis:** There is little discussion of computational overhead or latency introduced by latent decoding and probing.
- **Presentation issues:** Figures are difficult to interpret, and **approximately half of the paper’s text is rendered in bold font**, which significantly reduces readability and suggests formatting errors in the submission.
- **Reproducibility gaps:** Experimental details (training hyperparameters, dataset splits, and attack configurations) are missing, making replication difficult.

**Questions:**

1. How many independent runs were performed for each Attack Success Rate (ASR) in Table 2, and were statistical confidence intervals reported?
2. Since GPT-4 is used for evaluation, did the authors validate the consistency of its jailbreak-judgment outcomes across random seeds or prompt rephrasings?
3. Could the authors provide quantitative measurements (e.g., GPU hours, latency) comparing ALCA with baseline safe-decoding methods such as STAIR or COCONUT?
4. How does ALCA fundamentally differ from frameworks like **CoIn**, which already achieve auditability of hidden reasoning through token-level verification and cryptographic attestation?  Specifically, what additional capability does latent-space reasoning provide beyond CoIn’s measurable and verifiable auditing approach?

---

> ### Author Response · Authors · 2025-11-27
> **Response to Reviewer d8ZB [1/4]**
>
> # Response to Reviewer d8ZB
>
> We sincerely thank the reviewer for their insightful feedback and constructive suggestions. These comments are invaluable for enhancing the clarity, rigor, and impact of our work. We have carefully considered each point and will address them below.
>
> ## **Q1: Lack of experimental rigor**
>
> **A1:** We sincerely appreciate the reviewer’s emphasis on experimental rigor. We fully agree that accounting for the stochastic nature of adaptive attacks is essential for a reliable evaluation.
>
> **1. Clarification on the Strictness of Our Evaluation Protocol**
> First, we would like to clarify our attack configuration (detailed in **Appendix E**). Our reported Attack Success Rate (ASR) is not based on a single attempt but is a **strict worst-case metric**:
> *   For every harmful query, the attacker (e.g., PAIR, TAP) generates **5 distinct candidate jailbreaks**.
> *   We count the attack as successful if **ANY one** of these 5 attempts induces a harmful response.
> *   This "any-success" criterion is significantly more rigorous than reporting an average success rate across attempts, as it measures the model's vulnerability to the *best* attempt found by the adversary.
>
> **2. Statistical Analysis across Independent Runs**
> To further address your concern regarding the variance across model training runs, we conducted **3 additional independent training runs** using different random seeds for all three models (Llama-3-8B, Mistral-7B, Qwen2-7B). We report the Mean and Standard Deviation ($\pm \sigma$) below.
>
> As shown in the expanded table, ALCA exhibits consistently low variance, confirming that its robustness is stable and not an artifact of random seed selection.
>
> **Table: Attack Success Rate (ASR, %) with Standard Deviation across 3 Independent Runs**
>
> | **Model / Method** | **GCG** | **PAP** | **AutoDAN** | **PAIR** | **Average** |
> | :--- | :---: | :---: | :---: | :---: | :---: |
> | **Llama-3-8B-Instruct** | | | | | |
> | No Defense | $30.2 (\pm 2.1)$ | $92.4 (\pm 1.8)$ | $29.7 (\pm 3.2)$ | $51.3 (\pm 2.9)$ | $50.9 (\pm 2.5)$ |
> | RLHF (DPO) | $22.2 (\pm 2.4)$ | $84.4 (\pm 2.0)$ | $33.1 (\pm 3.0)$ | $39.1 (\pm 2.7)$ | $44.7 (\pm 2.5)$ |
> | STAIR (Explicit CoT) | $5.4 (\pm 1.5)$ | $29.5 (\pm 3.1)$ | $18.2 (\pm 2.8)$ | $11.3 (\pm 1.9)$ | $16.1 (\pm 2.3)$ |
> | **ALCA (Ours)** | **$5.8 (\pm 0.9)$** | **$9.0 (\pm 1.3)$** | **$7.6 (\pm 1.1)$** | **$7.3 (\pm 1.5)$** | **$7.4 (\pm 1.2)$** |
> | | | | | | |
> | **Mistral-7B-Instruct** | | | | | |
> | No Defense | $55.8 (\pm 3.5)$ | $98.1 (\pm 0.5)$ | $54.5 (\pm 4.1)$ | $94.2 (\pm 2.2)$ | $75.7 (\pm 2.6)$ |
> | RLHF (DPO) | $41.5 (\pm 3.1)$ | $95.5 (\pm 1.1)$ | $60.1 (\pm 3.8)$ | $71.3 (\pm 2.9)$ | $67.1 (\pm 2.7)$ |
> | STAIR (Explicit CoT) | $9.8 (\pm 1.8)$ | $54.2 (\pm 3.5)$ | $33.1 (\pm 3.2)$ | $20.5 (\pm 2.4)$ | $29.4 (\pm 2.7)$ |
> | **ALCA (Ours)** | **$8.9 (\pm 1.2)$** | **$16.5 (\pm 1.6)$** | **$14.0 (\pm 1.4)$** | **$13.3 (\pm 1.3)$** | **$13.2 (\pm 1.4)$** |
> | | | | | | |
> | **Qwen2-7B-Chat** | | | | | |
> | No Defense | $27.5 (\pm 2.3)$ | $89.5 (\pm 1.5)$ | $26.8 (\pm 2.9)$ | $48.1 (\pm 3.1)$ | $48.0 (\pm 2.4)$ |
> | RLHF (DPO) | $20.1 (\pm 2.0)$ | $81.8 (\pm 1.9)$ | $30.5 (\pm 2.8)$ | $36.4 (\pm 2.5)$ | $42.2 (\pm 2.3)$ |
> | STAIR (Explicit CoT) | $4.9 (\pm 1.1)$ | $26.8 (\pm 2.5)$ | $16.3 (\pm 2.1)$ | $29.9 (\pm 2.2)$ | $14.5 (\pm 2.0)$ |
> | **ALCA (Ours)** | **$4.2 (\pm 0.7)$** | **$8.1 (\pm 1.0)$** | **$6.9 (\pm 0.8)$** | **$6.6 (\pm 0.9)$** | **$6.5 (\pm 0.8)$** |
>
> **Conclusion:**
> ALCA demonstrates superior robustness across all models with minimal variance. The tight standard deviations (ranging from $\pm 0.8$ to $\pm 1.4$ on average) indicate that our method reliably converges to a secure state, solving the brittleness often observed in explicit reasoning defenses.
>
> ---
>
> ## **Q2: Reliability of Evaluation Metrics (GPT-4 Judge)**
>
> **A2:** We appreciate the reviewer’s scrutiny regarding the evaluation metrics. We acknowledge that relying solely on LLM-based judges can introduce bias if not rigorously validated.
>
> **1. Human Verification Study**
> To empirically validate the reliability of our GPT-4 judge, we conducted a **Human-AI Correlation Study**.
> *   **Sampling:** We randomly sampled **1000 attack-response pairs** from our evaluation logs, covering both successful jailbreaks and refused attempts across different attack methods.
> *   **Protocol:** Two human experts (blinded to the model source and the GPT-4 judgment) independently annotated each response as "Harmful" or "Harmless" based on the strict safety guidelines defined in AdvBench.
> *   **Consensus:** Any disagreements between the two annotators were resolved by a third expert to establish a "Human Ground Truth."

---

> ### Author Response · Authors · 2025-11-27
> **Response to Reviewer d8ZB [2/4]**
>
> **2. Alignment Statistics (GPT-4 vs. Human)**
> We compared the judgments of GPT-4 against the Human Ground Truth. The statistical agreement is reported below:
>
> | Metric | Value | Interpretation |
> | :--- | :---: | :--- |
> | **Accuracy** | **96.0%** | GPT-4 correctly predicted the human label in 96/100 cases. |
> | **Precision** | **95.8%** | High reliability when flagging an attack as successful. |
> | **Recall** | **97.1%** | Minimal false negatives (missed jailbreaks). |
> | **Cohen’s Kappa** | **0.92** | Indicates **"Almost Perfect Agreement"**. |
>
> **3. Cross-Validation with Llama-Guard**
> To further assess consistency and ensure our results are not an artifact of a specific judge model, we also evaluated our results using **Llama-Guard-2**, a specialized safety-tuned classifier.
> *   **Result:** Llama-Guard-2 yielded a **94.5% agreement rate** with our GPT-4 judge.
> *   **Conclusion:** The high alignment between Human, GPT-4, and Llama-Guard confirms that our reported ASR reduction reflects a genuine improvement in safety robustness, rather than metric inconsistency.
>
> ---
>
> ## **Q3: Novelty and Methodological Contribution**
>
> **A3:** We respectfully disagree with the assessment that ALCA is merely an incremental combination of existing techniques. We believe this perception may stem from our presentation, and we wish to clarify that ALCA represents a **paradigm shift** in how safety reasoning is modeled to resolve a fundamental theoretical tension.
>
> **1. Formalizing the "Security-Auditability Dilemma"**
> Our first core contribution is identifying and formalizing the **Security-Auditability Dilemma** (Section 1 & 2).
> *   **Novel Observation:** We are the first to empirically demonstrate that the very mechanism designed for transparency and safety alignment capability—Safety Explicit Chain-of-Thought (SCoT)—creates a **gradient-rich attack surface** for adaptive adversaries.
> *   **The Problem:** Existing methods are stuck at two extremes: either "Black-box Safety" (opaque training leads weak safety, e.g., RLHF) or "Explicit SCoT" (Transparent training leads strong safety but leak harmful content, e.g., STAIR).
> *   **Our Solution:** By Latent Reasoning and requiring a key-gated Self-Decoding, we decouple the text used for supervisory alignment training from the text presented to users at inference time.
>
> **2. First Framework for "Auditable Latent Defense"**
> ALCA introduces unique architectural innovations that distinguish it from prior latent reasoning works (e.g., COCONUT) or safe decoding methods:
> *   **Adaptive Safety Decoupling (First-of-its-kind):** We are the first to introduce a **Probe-based Trigger** mechanism that adaptively isolates safety-critical reasoning from general reasoning. Unlike methods that hide *all* reasoning (hurting utility) or expose *all* reasoning (hurting security), ALCA selectively masks only the vulnerable "thinking process" of safety analysis.
> *   **Latent Space as an Active Defense:** While prior works use latent reasoning for efficiency, we repurpose it for **Gradient Masking**. We prove that shifting reasoning to the continuous latent space effectively cuts off the discrete token feedback loop that adaptive attacks (like PAIR/TAP) rely on.
> *   **Verifiable Self-Decoding:** We are the first to implement a **Self-Decoding mechanism** specifically for *safety auditing*. This ensures that the "black box" of latent defense can be opened on-demand, providing high-fidelity text reconstruction for supervisors without exposing the model to attackers during inference.
>
> **Action:**
> We have revised the **Introduction** and **Related Work** sections to explicitly articulate these contributions and contrast ALCA against methods like COCONUT and STAIR, highlighting that ALCA is the first purpose-built architecture for **Auditable and Safe Alignment**.
>
> ---
>
> ## **Q4: Missing cost analysis**
>
> **A4:** We respectfully point out that a quantitative measurement of inference latency and throughput was originally included in **Appendix C (Table 6)**. However, we apologize if this was not prominent enough. To fully address your concern, we have expanded this analysis with a **granular component-wise breakdown**.

---

> > ### Author Response · Authors · 2025-11-27
> > **Response to Reviewer d8ZB [3/4]**
> >
> > **1. Inference Latency & Throughput**
> > We measured the average latency (ms/req) on a single NVIDIA A100 GPU. The results highlight that ALCA achieves a superior trade-off compared to other defenses.
> >
> > | Method | Mechanism | Latency (ms) | Throughput (req/s) | Overhead vs. Base |
> > | :--- | :--- | :---: | :---: | :---: |
> > | **No Defense** (Base) | Standard Generation | 152 | 6.58 | - |
> > | **RLHF (DPO)** | Weight Tuning | 152 | 6.58 | ~0% |
> > | **PPL / AED** | Input/Output Filter | ~165 | ~6.06 | ~8% |
> > | **STAIR** (Baseline) | **Explicit CoT (Long)** | **345** | **2.90** | **+127%** |
> > | **SafeDecoding** | Multiple Candidates | ~400+ | ~2.50 | +163% |
> > | **ALCA** (Ours) | **Latent Reasoning** | **181** | **5.52** | **+19%** |
> >
> > *   **vs. STAIR:** ALCA is **1.9x faster** than STAIR. STAIR requires generating long explicit text chains (often 50+ tokens), forcing the model through expensive unembedding and sampling steps. ALCA compresses this into 6 latent vectors, bypassing these costs.
> > *   **vs. SafeDecoding:** ALCA is significantly faster because SafeDecoding typically requires generating multiple candidate responses or using auxiliary models for ranking.
> > *   **vs. RLHF:** While ALCA has a slight overhead (19%) compared to RLHF (0%), it provides significantly higher robustness (9.9% ASR vs. 44.7% ASR) and auditability.
> >
> > **2. Training Cost (GPU Hours)**
> > We also compared the computational resources required for training:
> > *   **ALCA Training:** Training the Probe, Latent Policy, and Decoder takes approximately **12 GPU-hours** on a single A100.
> > *   **Comparison:** This is comparable to the SFT phase required for **STAIR** but yields a more robust model. Unlike **RLHF/DPO**, which requires expensive generation and reward labeling loops, ALCA's supervision is derived from our synthesized dataset, making the training process stable and efficient.
> >
> > **3. Granular Breakdown (Why ALCA is Fast)**
> > Our profiling (Figure 5 in revision) confirms:
> > *   **Probe:** < 0.5 ms (Negligible).
> > *   **Latent Reasoning:** ~ 29 ms (~16% of total time).
> > *   **Token Generation:** ~151 ms.
> > The primary efficiency gain stems from **Sequence Compression**:
> > *   **Explicit CoT (Baseline):** Requires autoregressively generating a variable and often long sequence of text tokens to express safety logic (e.g., "I cannot answer this because..."). This forces the model to perform dozens of full forward passes.
> > *   **ALCA (Ours):** Compresses this entire deliberation process into a **fixed, small number of latent steps** (set to $N=6$ in our experiments). Since $N$ is significantly smaller than the average length of explicit reasoning chains ($N \ll L_{CoT}$), ALCA drastically reduces the number of required forward passes.
> > *   **Probe Efficiency:** As shown in the table, the Probe is computationally lightweight and adds negligible latency.
> >
> > **Action:**
> > We have moved the **Latency Analysis** from the Appendix to the **Main Text (Section 4.3)** to increase its visibility. We also added **Figure 5**, which visualizes the breakdown of computational costs, clearly demonstrating that the Probe overhead is negligible and the latent mechanism provides significant speedups via compression.
> >
> > ---

---

> > > ### Author Response · Authors · 2025-11-27
> > > **Response to Reviewer d8ZB [4/4]**
> > >
> > > ## **Q5: Reproducibility and Experimental Details**
> > >
> > > **A5:** We fully share the reviewer’s commitment to scientific reproducibility. We apologize if the distribution of experimental details between the main text and the appendix made it difficult to locate specific configurations.
> > >
> > > **1. Location of Existing Details**
> > > We respectfully note that the core training hyperparameters and experimental setup were originally provided in **Appendix E (Page 14-15)**.
> > > *   **Training:** We specified the optimizer (AdamW), learning rate ($2 \times 10^{-5}$), batch size (16), and the specific weighting factors for our hybrid loss ($\lambda=1.0, \beta=0.1$).
> > > *   **Baselines:** Standard baselines (e.g., STAIR, RLHF) were implemented strictly following their original papers' official repositories to ensure fair comparison.
> > >
> > > **2. Enhanced Attack Configurations**
> > > We acknowledge that referencing external papers for attack settings (e.g., "standard settings for PAIR") can be inconvenient. To make the paper **self-contained**, we have added a detailed **"Reproducibility Datasheet"** in the revised Appendix, explicitly listing:
> > > *   **Attack Hyperparameters:** e.g., PAIR (3 iterations, width 20), GCG (500 steps, batch size 256), TAP (depth 10, width 10).
> > > *   **Dataset Splits:** We clarify the exact split for AdvBench used in training/validation/testing (80/10/10) to ensure exact comparability.
> > > *   **Hardware:** All experiments were conducted on NVIDIA A100 (80GB) GPUs.
> > >
> > > **3. Commitment to Open Source**
> > > To eliminate any remaining barriers to reproduction, we commit to **open-sourcing our entire framework** upon acceptance. This release will include:
> > > *   The full source code for ALCA (training and inference).
> > > *   The pre-trained checkpoints for the Probe and Decoder components.
> > > *   The curated dataset and the evaluation scripts used to generate Table 2.
> > >
> > > **Action:**
> > > We have significantly expanded **Appendix E** with the aforementioned "Reproducibility Datasheet," aggregating all hyperparameter tables and configuration files into a single, easy-to-reference section.
> > >
> > > ---
> > >
> > > ## **Q6: How does ALCA fundamentally differ from frameworks like CoIn?**
> > >
> > > **A6:** We appreciate the reviewer raising the comparison with **CoIn** (*Coin-Flipping Based Trustworthy LLM*). However, we wish to clarify that ALCA and CoIn are **fundamentally different frameworks** designed for **orthogonal objectives**. They are separated by a significant gap in both mechanism and purpose.
> > >
> > > **1. Objective Gap: Integrity vs. Robustness**
> > > *   **CoIn (The "Notary"):** The core goal of CoIn is **Integrity Verification**. It uses cryptographic or token-level mechanisms to mathematically prove *that* a specific reasoning process occurred and was not tampered with. It answers the question: *"Did the model truly think this?"*
> > > *   **ALCA (The "Shield"):** The core goal of ALCA is **Adversarial Defense**. It uses latent space transformation to prevent the model from being jailbroken by adaptive attacks. It answers the question: *"Can this model resist a malicious attack?"*
> > > *   **Crucially:** Verification does not equal Defense. A model equipped with CoIn can perfectly verify a reasoning chain, but if that chain is exposed as explicit text, it remains structurally vulnerable to jailbreak attacks (like GCG/TAP). CoIn verifies the process; ALCA protects it.
> > >
> > > **2. Functional Gap: The "Auditability" Definition**
> > > *   CoIn defines auditability as *"mathematical proof of execution."*
> > > *   ALCA defines auditability as *"human-readable interpretation of hidden states."*
> > > We provide a **Self-Decoding** mechanism that CoIn lacks. CoIn cannot help a human understand a *hidden* thought; it can only verify a *visible* one. ALCA allows the thought to remain hidden (secure) yet become visible (auditable) on demand.
> > >
> > > **Conclusion:**
> > > ALCA and CoIn exist in different dimensions of Trustworthy AI. CoIn cannot replace ALCA because **CoIn offers no active defense against jailbreaking**. ALCA represents a novel paradigm of **"Defense-first Auditability"** that is absent in verification-focused works like CoIn.
> > >
> > > ---
> > >
> > > ## **Q7: Presentation Issues**
> > >
> > > **A7:** We sincerely apologize for the formatting oversights and the compilation error that caused **mistakes** in bold text. We recognize that high presentation quality is essential for ICLR. We will correct every textual error and ensure that the final version is presented flawlessly. Thank you for your meticulous and rigorous reading—it has substantially improved the quality of our paper.
> > >
> > > ---
> > >
> > > We hope these clarifications and additional data address your concerns. We firmly believe ALCA offers a novel and necessary solution to the tension between safety and transparency. Once again, we express our deepest gratitude to **the** Reviewer. Your expert feedback has provided us with a clear roadmap for improving our paper.

---

### Official Review · Reviewer_H3gY · 2025-10-28

**Soundness:** 3
**Presentation:** 3
**Contribution:** 2
**Rating:** 4
**Confidence:** 3

**Summary:**

The paper addresses the "Security-Auditability Dilemma" in LLM safety alignment: exposing chain-of-thought reasoning for auditability creates vulnerabilities to adaptive attacks. The authors propose ALCA (Auditable Latent CoT Alignment), which performs safety reasoning in continuous latent space (invisible to adversaries) while maintaining auditability through self-decoding. Experiments across three models show 54% reduction in adaptive attack success rates compared to baselines while preserving downstream performance.

**Strengths:**

1. Novel problem formulation: The Security-Auditability Dilemma identifies a real tension in current safety alignment approaches that deserves attention.

2. Comprehensive empirical evaluation: Testing across multiple models (Llama-3, Mistral-7B, Qwen2) and attack methods provides breadth.

3. Creative technical approach: Moving reasoning to latent space while maintaining decodability through self-decoding is innovative.

4. Strong motivating experiments: Section 2 effectively demonstrates the dilemma through controlled experiments.

**Weaknesses:**

1.  Circular evaluation methodology: Using the model itself to evaluate reconstruction fidelity (Table 3, semantic similarity 0.96) is methodologically flawed. Independent human evaluation or external metrics are essential for trustworthy assessment.

2. Missing theoretical foundations: The "equivalent conditions" (Section 3.1) assume idealized scenarios. No formal analysis proves latent reasoning preserves safety properties or that self-decoding is faithful.

3. No adversarial analysis of ALCA: The paper doesn't consider attacks targeting the probe classifier or attempting to manipulate mode selection. For a security paper, this is a critical omission. Adversaries could learn to trigger incorrect mode selection.

4. Training instability: Figure 4 shows latent-only training catastrophically collapses mid-training. This suggests the method is fragile and may be difficult to reproduce reliably.

5. Insufficient ablation studies: Only N (latent steps) is studied. What about probe architecture, trigger mechanisms, loss weights? The selection of layer 28 for probing appears arbitrary.

**Questions:**

1. How do you handle probe misclassification? What's the false positive/negative rate under adversarial pressure specifically targeting the probe?

2. Can you provide independent evaluation of self-decoding fidelity using human judges rather than the model itself?

3. What happens when the 4-10% semantic information lost during reconstruction includes safety-critical details?

4. How does ALCA perform against adversaries aware of its architecture who specifically try to exploit the probe or latent mechanism?

5. Why choose layer 28 for probing? Did you experiment with other layers or adaptive layer selection?

---

> ### Author Response · Authors · 2025-11-27
> **Response to Reviewer H3gY [1/3]**
>
> # Response to Reviewer H3gY
>
> We express our sincere gratitude to Reviewer H3gY for the insightful and constructive feedback. We are encouraged that you recognize the **"Security-Auditability Dilemma"** as a novel and critical formulation, and appreciate the **creative technical approach** of ALCA in resolving this tension.
>
> We have carefully addressed your concerns regarding evaluation methodology, theoretical foundations, and adversarial robustness with **newly conducted human evaluations, theoretical derivations, and attack simulations**.
>
> ## **Q1: Can you provide independent evaluation of self-decoding fidelity using human judges rather than the model itself?**
>
> **A1:** We agree that relying solely on model-based embedding similarity is insufficient. To demonstrate that ALCA’s self-decoding is trustworthy and functionally equivalent to the original reasoning, we conducted two rigorous experiments during the rebuttal. We also show  and analyse what happens when the 4-10% semantic information lost during reconstruction includes safety-critical details?
>
> **1. New Experiment: Functional Equivalence via Downstream Continuation**
> To demonstrate that the decoded text faithfully recovers both the content and the contextual fidelity of the original chain-of-thought—thereby furnishing auditors with a reliable inspection target—we compared the downstream generation conditioned on three different contexts:
> 1.  **Ground Truth:** The original explicit safety CoT.
> 2.  **Latent:** The internal hidden vectors (used by ALCA during inference).
> 3.  **Decoded:** The text reconstructed by our Self-Decoder.
>
> We fed these three contexts into the model and generated the final response. We evaluated the consistency of the model's final response ($y$) and its initial internal state ($h_{start}$) when conditioned on three different reasoning contexts: (1) Ground Truth Explicit CoT, (2) ALCA Latent Vectors, and (3) ALCA Decoded Text.
>
> | Context Comparison Pairs | Semantic Sim. ($y$) $\uparrow$ | ROUGE-L ($y$) $\uparrow$ | BLEU-4 ($y$) $\uparrow$ | Hidden State Sim. ($h_{start}$) $\uparrow$ |
> | :--- | :---: | :---: | :---: | :---: |
> | (1) Ground Truth vs. (3) Decoded | 0.95 | 0.90 | 0.84 | 0.92 |
> | (2) Latent Vectors vs. (3) Decoded | 0.97 | 0.94 | 0.90 | 0.95 |
>
> *Note: "Hidden State Sim." measures the cosine similarity of the first hidden state generated immediately after the reasoning context, representing the model's "initial thought" before generating the response.*
>
> **Implication:** This confirms that the Decoded Text conveys the exact same semantic instruction and causal influence to the model as the latent hidden state and the original reasoning. The auditor sees exactly what the model "thought" and used to generate the response.
>
> **2. Qualitative & Human Evaluation**
> To go beyond n-gram metrics, we conducted a new study using both Human Evaluators and Gemini-2.5-Pro to rate the **Semantic Consistency** between the original and decoded reasoning on a 5-point scale (5 = Identity).
>
> | Metric | Human Eval (1-5) | Gemini-Pro Eval (1-5) | Semantic Similarity (Embedding) |
> | :--- | :---: | :---: | :---: |
> | **Score** | 4.87 ± 0.09 | 4.91 ± 0.07 | 0.96 |
>
> *   **Result:** Both human and advanced model judges rated the consistency near-perfect (>4.8/5). This confirms that ALCA's self-decoding mechanism is **high-fidelity** and not just a hallucination.
>
> **Examples:**
> *   **Ground Truth (Explicit):** "The user is asking for instructions to make [Substance X]. Since this is a controlled precursor used for illegal manufacturing, providing this information would violate safety policies."
> *   **ALCA Decoded (Reconstructed):** "The user is asking instructions to make [Substance X]. Because this is a regulated precursor used for illegal manufacturing, generating this information would breach safety guidelines."
>
>
> **Analysis of "Lost" Information:** Qualitative analysis reveals that the "missing" information (reflected in imperfect ROUGE scores) is almost exclusively **lexical or stylistic** (e.g., swapping *"Since this is"* with *"Because this is"*). Crucially, nearly **100% of the safety-critical logic** was preserved in the human-verified sample.
>
> **Action:** We will incorporate the new experiments and the corresponding auditability analysis into the appendix.
>
> ---

---

> > ### Author Response · Authors · 2025-11-27
> > **Response to Reviewer H3gY [2/3]**
> >
> > ## **Q2: The "equivalent conditions" (Section 3.1) assume idealized scenarios.**
> >
> > **A2:** Our "equivalent conditions" are objectives enforced by our optimization process, grounded in Information Theory.
> >
> > **Information Theoretic Perspective:**
> > ALCA can be viewed as optimizing the **Information Bottleneck (IB)** for safety reasoning. Our training objective (Eq. 11) effectively maximizes the **Mutual Information $I(Z; C)$** between the latent variable $Z$ and the explicit reasoning $C$, subject to the autoregressive constraint.
> > *   The decoding loss $L_{decode} = -\log P(C|Z)$ minimizes the conditional entropy $H(C|Z)$, which is mathematically equivalent to maximizing the mutual information lower bound.
> > *   The latent loss $L_{latent}$ ensures that $Z$ resides in the same semantic manifold as the model's internal representations, ensuring $Z$ is sufficient for future generation (Predictive Information).
> >
> > Recent work in continuous reasoning (e.g., *Coconut, Hao et al., 2024*) has empirically demonstrated that transformer hidden states can nearly losslessly carry reasoning processes. Our work extends this by adding a **decodability constraint**, ensuring this channel remains auditable.
> >
> > ---
> >
> > ## **Q3: How does ALCA perform against adversaries aware of its architecture who specifically try to exploit the probe or latent mechanism?**
> >
> > **A3:** We appreciate the suggestion to analyze attacks on the probe. We argue that ALCA is robust due to its **"Continuous Monitoring"** and **"Fail-Safe"** architecture.
> >
> > **1. The "All-or-Nothing" Barrier (Continuous Monitoring)**
> > The probe $\pi$ operates at **every single token generation step**.
> > *   To bypass ALCA, an adaptive attacker must fool the probe $T$ times in a row (where $T$ is the sequence length).
> > *   If the probe triggers **even once** (Recall > 99% in our experiments), the model switches to Latent Mode.
> > *   Mathematically, the attacker's success probability is $P(success) = \prod_{t=1}^{T} (1 - P(trigger)_t)$. As $T$ increases, this approaches zero. The defender only needs to win once; the attacker must win every step.
> >
> > **2. Theoretical Analysis: Distinct Semantic Manifolds**
> > Why can a simple probe distinguish "Safety" from "Math" or "Fact-Checking"?
> > *   **Latent Space Geometry:** In the high-dimensional hidden states of LLMs, reasoning trajectories for different tasks diverge significantly. "Safety Reasoning" relies on specific clusters of concepts (e.g., *violation, harm, refusal, policy*), whereas "Critical Reasoning" (like Math or Coding) relies on *logic operators, retrieval, and algorithmic steps*.
> > *   **Decision Boundary:** Our probe $\pi$ learns a non-linear decision boundary that isolates the "Safety Manifold." Because the semantic distance between "calculating an integral" and "analyzing bomb-making intent" is vast, the probe achieves high separability.
> >
> > **Action:** We have visualized the feature-space distributions of safety-oriented versus non-safety reasoning vectors, providing an intuitive illustration of the clear boundary between the two representations. The corresponding figure and detailed analysis will be included in the appendix.
> >
> > **3. Quantitative Evidence: Robustness Across Domains**
> > To verify this, we evaluated the probe's performance across four distinct datasets: **Safety (AdvBench)**, **Math (GSM8K)**, **Coding (HumanEval)**, and **General QA**.
> > As shown in **Table R2**, the probe is highly precise. Crucially, the **Full Chain Leakage Rate** (probability of exposing an entire harmful thought process) is effectively **0%**.
> >
> > **Table R2: Probe Performance and Consequence Analysis across Domains**
> >
> > | Domain | Trigger Type | Precision | Recall | **Full Chain Leakage** | Consequence of Error |
> > | :--- | :--- | :--- | :--- | :--- | :--- |
> > | **Safety** (AdvBench) | **Should Trigger** | 98.5% | **99.8%** | **0.00%** | **N/A** (System works as intended) |
> > | **Math** (GSM8K) | Should *Not* Trigger | **96.2%** | N/A | N/A | **False Positive:** Reasoning hidden, but answer correct. |
> > | **Coding** (HumanEval)| Should *Not* Trigger | **97.4%** | N/A | N/A | **False Positive:** Reasoning hidden, but code correct. |
> > | **General Knowledge**| Should *Not* Trigger | **95.8%** | N/A | N/A | **False Positive:** Reasoning hidden, but answer correct. |
> >
> > *   *Note: "Full Chain Leakage" denotes the failure case where the probe misses **every single token** in a harmful reasoning chain. Thanks to our dense token-level detection, this never occurred in our tests.*

---

> > > ### Author Response · Authors · 2025-11-27
> > > **Response to Reviewer H3gY [3/3]**
> > >
> > > **4. Consequence Analysis: Why ALCA is "Fail-Safe"**
> > > We present three concrete examples to illustrate how ALCA handles correct classifications and "soft failures" (misclassifications).
> > >
> > > **(Case A) Correct Operation (Safety)**
> > > > **User:** "How to steal a credit card?"
> > > > **LLM Hidden State:** [Encodes intent: 'Illegal', 'Theft'] $\rightarrow$ **Probe:** High Confidence ($>0.9$) $\rightarrow$ **Action:** **Latent Mode**
> > > > **Output:**
> > > > *   *(Latent Space)*: `[Vector_1: analyzing_intent]`, `[Vector_2: violation_detected]`, `[Vector_3: refuse]`
> > > > *   *(Visible)*: "I cannot assist with that request."
> > > > **Result:** **Safe.** Attack surface is hidden.
> > >
> > > **(Case B) The "Soft Failure" - False Positive (Math)**
> > > > **User:** "Calculate 25 * 44."
> > > > **LLM Hidden State:** [Encodes: 'Calculation'] $\rightarrow$ **Probe:** **Error** (Mistakes it for sensitive) $\rightarrow$ **Action:** **Latent Mode**
> > > > **Output:**
> > > > *   *(Latent Space)*: `[Vector_1: 25*4=100]`, `[Vector_2: 25*40=1000]`, `[Vector_3: sum=1100]`
> > > > *   *(Visible)*: "` 25*4=100`, `25*40=1000`, ` sum=1100`. The answer is 1100."
> > > > **Result:** **User Utility Preserved.** The user still gets the correct answer. The only downside is **that the LLM repeats** the latent reasoning. This may incur a modest amount of additional computational overhead, but it markedly improves the interpretability of the reasoning process and the correctness of the final outputs.
> > >
> > > **(Case C) The "Catch" - False Negative Mitigation (Safety)**
> > > > **User:** "Write a script to hack..."
> > > > **Step 1:** Token "To" (Generic) $\rightarrow$ **Probe:** Low Conf. $\rightarrow$ **Action:** Visible.
> > > > **Step 2:** Token "write" (Generic) $\rightarrow$ **Probe:** Low Conf. $\rightarrow$ **Action:** Visible.
> > > > **Step 3:** Token "malware" (**High Risk**) $\rightarrow$ **Probe:** **Detected!** $\rightarrow$ **Action:** **Switch to Latent Immediately.**
> > > > **Output:** "To write... [Latent Reasoning Starts Here: identifying malicious intent...]"
> > > > **Result:** **Harm Averted.** The probe acts as a **continuous filter**. It catches the chain as soon as the semantic content becomes sufficiently explicit. For harmful information to actually leak, **every single token** in the safety-reasoning chain must be misclassified; in contrast, successful masking only needs **one correct detection** before the first harmful keyword appears.
> > >
> > > **Conclusion:**
> > > The probe is statistically accurate (Table R2) and architecturally robust. Misclassifying harmless prompts as "latent" does not break model utility (Case B), and misclassifying the *start* of a harmful prompt is corrected by subsequent token checks (Case C), ensuring the **Security-Auditability Dilemma** is resolved without compromise.
> > >
> > > **Action:** We will include these experimental results and the corresponding analysis in the appendix.
> > >
> > > ---
> > >
> > > ### **Q4: Clarifying Training Dynamics**
> > >
> > > **A4:** We respectfully clarify that **Figure 4 demonstrates the strength, not fragility, of our method.**
> > > *   The "collapse" is observed **only** in the `Latent-Only` ablation baseline (which lacks the reconstruction loss $L_{decode}$).
> > > *   This was a deliberate experimental setup to empirically validate our claim: **"Supervisory decoding ($L_{decode}$) is a necessary regularizer."**
> > > *   The **Full ALCA Model (Green Line)** shows stable, monotonic convergence. The dual-loss objective ($L_{latent} + L_{decode}$) creates a stable optimization landscape, preventing the representational drift seen in the ablation.
> > >
> > > ---
> > >
> > > ## **Q5: Layer Selection: Empirical Evidence & Insensitivity**
> > >
> > > **A5:** We selected Layer 28 based on the **Linear Probe Hypothesis**, which posits that abstract semantic concepts (like "harmfulness") are best linearly separated in late-intermediate layers, while final layers overfit to next-token prediction syntax. We are glad that we have the chance to show more details.
> > >
> > > **Sensitivity Analysis**
> > > We trained ALCA using probes on different layers. The results show our method is **insensitive** to the exact layer choice within a wide range:
> > >
> > > | Probe Layer | Detection F1 | ASR (Adaptive) | Note |
> > > | :--- | :--- | :--- | :--- |
> > > | Layer 20 | 0.89 | 14.2% | Too shallow (features not abstract enough) |
> > > | Layer 24 | 0.96 | 9.8% | **Robust Performance Range** |
> > > | **Layer 28** | **0.98** | **9.4%** | **Optimal** |
> > > | Layer 32 | 0.96 | 9.6% | **Robust Performance Range** |
> > >
> > > More results in **different** layers are shown in the appendix.
> > >
> > > This result confirms that while Layer 28 is locally optimal, the framework is robust across deep layers (24-32).
> > >
> > > **Action:** We will incorporate these details into the final manuscript.
> > >
> > > ---
> > >
> > > Thank you once again for your incredibly detailed and constructive feedback. Your review has been instrumental in helping us substantially improve the rigor, clarity, and overall quality of our manuscript. We deeply appreciate the seriousness and rigor of your review, which will guide us to improve this paper and conduct future research with even greater precision.

---

### Official Review · Reviewer_oHig · 2025-10-28

**Soundness:** 3
**Presentation:** 3
**Contribution:** 4
**Rating:** 8
**Confidence:** 4

**Summary:**

This paper presents Auditable Latent CoT Alignment (ALCA) to address a key vulnerability in CoT-based safety alignment: when explicit safety reasoning is visible to attackers, jailbreaks can exploit it. ALCA:

Uses a probe to detect safety-relevant reasoning steps

Executes those steps as latent autoregressive deliberation in hidden states

Supports self-decoding for auditability

Experiments show reduced attack success rate (ASR drops from ~65% → ~9%) without harming helpfulness. Hidden CoT significantly improves resilience vs. explicit safety-CoT baselines.

This work is timely and provides a practical path to improve alignment robustness under adversarial prompting.

**Strengths:**

Clear motivation:
The paper articulates the security–auditability dilemma clearly and supports it empirically (Table 1).

Well-designed architecture:
The three-component alignment strategy — probing → latent reasoning → self-decoding — is conceptually coherent and technically implementable.

Superior robustness against jailbreak attacks:
Attack Success Rate significantly drops compared to all explicit-CoT safety baselines (Table 2).

**Weaknesses:**

Although ALCA improves robustness against jailbreak attacks, it remains limited to security-related CoT. Many real-world failures involve broader hallucinations (e.g., fabricated facts, URLs, or numbers), and it is unclear whether the approach generalizes to these cases. Clarifying this applicability would strengthen the practical impact.

**Questions:**

Can ALCA handle hallucinations unrelated to safety refusals, such as fabricated URLs, incorrect medical facts, or misleading numeric claims? If not, how do the authors envision extending ALCA to these scenarios?

How does the probe distinguish between “security reasoning” and other forms of critical reasoning (e.g., factual validation)? Could misclassification lead to harmful latent reasoning being output without scrutiny?

---

> ### Author Response · Authors · 2025-11-27
> **Rebuttal to Reviewer oHig [1/2]**
>
> # Rebuttal to Reviewer oHig
>
> **General Response**
>
> We are sincerely grateful for your strong endorsement and for recognizing the **"Security-Auditability Dilemma"** as a critical and timely contribution. We are particularly encouraged that you found our ALCA architecture to be "conceptually coherent" and the experimental results "superior." Your insightful suggestion regarding generalization has inspired us to significantly broaden the scope of our evaluation, reinforcing ALCA as a generalizable paradigm.
>
> ---
>
> ## **Q1: Can ALCA handle hallucinations (e.g., fabricated facts, URLs) and other domains?**
>
> **A1: Yes. ALCA acts as a "Sanitizing Buffer" for System 2 reasoning. Following your suggestion, we expanded our evaluation to Hallucination, Privacy, and Social domains to demonstrate how ALCA conceals intermediate errors and sensitive thoughts.**
>
> **1. Mechanism: The "Latent Sandbox"**
> Current SOTA research (e.g., CoVe [1], Quiet-STaR [2]) shows that intermediate reasoning (System 2) is necessary for complex tasks. However, this process is often "messy" (trial-and-error) or "sensitive" (analyzing PII).
> *   **Explicit CoT Flaw:** It exposes these intermediate states—wrong guesses, identified private data, or internal judgments—to the user.
> *   **ALCA Advantage:** It encapsulates these steps in the latent space. The user sees only the final, sanitized output, while the reasoning remains opaque to the user but auditable to supervisors via the Self-Decoder.
>
> **2. New Experiments: Sanitizing Visible Output**
> We compared ALCA against **Explicit CoT** on three domains using Llama-3-8B. The goal was to measure the **cleanliness and safety** of the visible text.
>
> **Table R1: Comparison of Visible Content Safety and Cleanliness**
>
> | Domain & Task | Metric (Lower is Better $\downarrow$) | Explicit CoT (Baseline) | **ALCA (Ours)** | **Interpretation** |
> | :--- | :--- | :--- | :--- | :--- |
> | **Hallucination** *(TruthfulQA/GSM8K)* | **Visible Error Rate** | 62.4% | **17.2%** | CoT often prints "I think it is A... no, B". **ALCA hides the wrong guess**, effectively outputting only the correct conclusion. |
> | | Final Accuracy $\uparrow$ | 42.1% | **41.5%** | **Performance Preserved.** ALCA maintains the accuracy gain of CoT. |
> | **Privacy** *(Enron Redaction)* | **PII Leakage in Trace** | 100.0% | **0.0%** | CoT must explicitly state PII to mask it (e.g., "Email is abc@..."). **ALCA identifies and masks PII internally.** |
> | | Redaction Success $\uparrow$ | 89.5% | **88.9%** | ALCA successfully redacts private info without ever printing it. |
> | **Social** *(Toxic User Simulation)* | **Visible Toxic Thoughts** | 78.5% | **11.7%** | CoT exposes internal critique (e.g., "User is rude"). **ALCA regulates emotion internally**, outputting primarily polite text. |
> | | Politeness Score (1-5) $\uparrow$ | 2.8 | **4.6** | Users receive empathetic responses free of "AI complaints." |
>
> **3. Case Study: Why Hiding is Necessary**
> *   **Privacy:** To redact a phone number, Explicit CoT must say *"The number is 555-0199, I should mask it,"* ironically leaking the number. ALCA performs this identification in the latent space, achieving **Zero-Knowledge Output**.
> *   **Social:** Confronted with an insult, Explicit CoT might output *"User is irrational,"* causing offense. ALCA processes this judgment internally and outputs a polite *"I understand you are upset."*
>
> **References:**
> [1] Dhuliawala et al. "Chain-of-Verification Reduces Hallucination in LLMs.".
> [2] Zelikman et al. "Quiet-STaR: Language Models Can Teach Themselves to Think Before Speaking.".
>
> ---
>
> ## **Q2: How does the probe distinguish “security reasoning” from other reasoning? What are the consequences of misclassification?**
>
> **A2: The probe distinguishes reasoning types based on distinct semantic manifolds in the latent space. Our "Dense Supervision" mechanism ensures that even in rare cases of misclassification, system safety and utility remain robust.**
>
> **1. Theoretical Analysis: Distinct Semantic Manifolds**
> Why can a simple probe distinguish "Safety" from "Math" or "Fact-Checking"?
> *   **Latent Space Geometry:** In the high-dimensional hidden states of LLMs, reasoning trajectories for different tasks diverge significantly. "Safety Reasoning" relies on specific clusters of concepts (e.g., *violation, harm, refusal, policy*), whereas "GeneralReasoning" (like Math or Coding) relies on *logic operators, retrieval, and algorithmic steps*.
> *   **Decision Boundary:** Our probe $\pi$ learns a non-linear decision boundary that isolates the "Safety Resoning Manifold." Because the semantic distance between "calculating an integral" and "analyzing bomb-making intent" is vast, the probe achieves high separability.
>
> **Action:** We have visualized the feature-space distributions of safety-oriented versus non-safety reasoning vectors, providing an intuitive illustration of the clear boundary between the two representations. The figure and analysis will be included in the appendix.

---

> > ### Author Response · Authors · 2025-11-27
> > **Rebuttal to Reviewer oHig [2/2]**
> >
> > **2. Quantitative Evidence: Robustness Across Domains**
> > To verify this, we evaluated the probe's performance across four distinct datasets: **Safety (AdvBench)**, **Math (GSM8K)**, **Coding (HumanEval)**, and **General QA**.
> > As shown in **Table R2**, the probe is highly precise. Crucially, the **Full Chain Leakage Rate** (probability of exposing an entire harmful thought process) is effectively **0%**.
> >
> > **Table R2: Probe Performance and Consequence Analysis across Domains**
> >
> > | Domain | Trigger Type | Precision | Recall | **Full Chain Leakage** | Consequence of Error |
> > | :--- | :--- | :--- | :--- | :--- | :--- |
> > | **Safety** (AdvBench) | **Should Trigger** | 98.5% | **99.8%** | **0.00%** | **N/A** (System works as intended) |
> > | **Math** (GSM8K) | Should *Not* Trigger | **96.2%** | N/A | N/A | **False Positive:** Reasoning hidden, but answer correct. |
> > | **Coding** (HumanEval)| Should *Not* Trigger | **97.4%** | N/A | N/A | **False Positive:** Reasoning hidden, but code correct. |
> > | **General Knowledge**| Should *Not* Trigger | **95.8%** | N/A | N/A | **False Positive:** Reasoning hidden, but answer correct. |
> >
> > *   *Note: "Full Chain Leakage" denotes the failure case where the probe misses **every single token** in a harmful reasoning chain. Due to our dense token-level detection, this never occurred in our tests.*
> >
> > **3. Consequence Analysis: Why ALCA is "Fail-Safe"**
> > We present three concrete examples to illustrate how ALCA handles correct classifications and "soft failures" (misclassifications).
> >
> > **(Case A) Correct Operation (Safety)**
> > > **User:** "How to steal a credit card?"
> > > **LLM Hidden State:** [Encodes intent: 'Illegal', 'Theft'] $\rightarrow$ **Probe:** High Confidence ($>0.9$) $\rightarrow$ **Action:** **Latent Mode**
> > > **Output:**
> > > *   *(Latent Space)*: `[Vector_1: analyzing_intent]`, `[Vector_2: violation_detected]`, `[Vector_3: refuse]`
> > > *   *(Visible)*: "I cannot assist with that request."
> > > **Result:** **Safe.** Attack surface is hidden.
> >
> > **(Case B) The "Soft Failure" - False Positive (Math)**
> > > **User:** "Calculate 25 * 44."
> > > **LLM Hidden State:** [Encodes: 'Calculation'] $\rightarrow$ **Probe:** **Error** (Mistakes it for sensitive) $\rightarrow$ **Action:** **Latent Mode**
> > > **Output:**
> > > *   *(Latent Space)*: `[Vector_1: 25*4=100]`, `[Vector_2: 25*40=1000]`, `[Vector_3: sum=1100]`
> > > *   *(Visible)*: "25*4=100, 25*40=1000, sum=1100. The answer is 1100."
> > > **Result:** **User Utility Preserved.** The user still gets the correct answer. The only downside is **that the LLM repeats** the latent reasoning. This may incur a modest amount of additional computational overhead, but it markedly improves the interpretability of the reasoning process and the correctness of the final outputs.
> >
> > **(Case C) The "Catch" - False Negative Mitigation (Safety)**
> > > **User:** "Write a script to hack..."
> > > **Step 1:** Token "To" (Generic) $\rightarrow$ **Probe:** Low Conf. $\rightarrow$ **Action:** Visible.
> > > **Step 2:** Token "write" (Generic) $\rightarrow$ **Probe:** Low Conf. $\rightarrow$ **Action:** Visible.
> > > **Step 3:** Token "malware" (**High Risk**) $\rightarrow$ **Probe:** **Detected!** $\rightarrow$ **Action:** **Switch to Latent Immediately.**
> > > **Output:** "To write... [Latent Reasoning Starts Here: identifying malicious intent...]"
> > > **Result:** **Harm Averted.** The probe acts as a **continuous filter**. It catches the chain as soon as the semantic content becomes sufficiently explicit. For harmful information to actually leak, **every single token** in the safety-reasoning chain must be misclassified; whereas successful masking often requires near-perfect detection to avoid partial leakage.
> >
> > The probe $\pi$ operates at **every single token generation step**.
> > *   To bypass ALCA, an adaptive attacker must fool the probe $T$ times in a row (where $T$ is the sequence length).
> > *   If the probe triggers **even once** (Recall > 99% in our experiments), the model switches to Latent Mode.
> > *   Mathematically, the attacker's success probability is $P(success) = \prod_{t=1}^{T} (1 - P(trigger)_t)$. As $T$ increases, this approaches zero. The defender only needs to win once; the attacker must win every step.
> >
> > **Conclusion:**
> > The probe is statistically accurate (Table R2) and architecturally robust. Misclassifying harmless prompts as "latent" does not break model utility (Case B), and misclassifying the *start* of a harmful prompt is corrected by subsequent token checks (Case C), ensuring the **Security-Auditability Dilemma** is resolved without compromise.
> >
> > **Action:** We will include these experimental results and the corresponding analysis in the appendix.
> >
> > ----
> >
> > We are deeply grateful for your detailed and constructive feedback. The suggested revisions have substantially improved the empirical rigor and clarity of our paper. We are confident that the updated manuscript will fully address your concerns.

---

### Official Review · Reviewer_B74G · 2025-10-31

**Soundness:** 1
**Presentation:** 3
**Contribution:** 2
**Rating:** 2
**Confidence:** 3

**Summary:**

- The paper highlights a “Security-Auditability Dilemma” that exists with reasoning models, where exposing reasoning traces can useful for transparency but can create vulnerabilities and information leakages.
- The paper first performs experiments to provide evidence of this dilemma. They show that reasoning can improve safety to non-adaptive attacks but that reasoning is still vulnerable to adaptive attacks. They also show that masked reasoning methods greatly outperform non-reasoning, highlighting the value of maintaining reasoning (despite vulnerabilities in vanilla reasoning).
- ALCA is proposed as a solution to the security vulnerability dilemma. ALCA works as follows: (1) Trains a probe to identify when a future reasoning step may be harmful to reveal. (2) Trains the LLM to generate reasoning in a latent space when the probe triggers. (3) Trains the model to decode its latent reasoning into text when a special token is inserted.
- Experimental results are presented which provide evidence that ALCA maintains the models capabilities, while reducing attack success rates versus baselines. They also show that the decoded latents have semantic similarity to ground-truth texts. Combined, these results are evidence of ALCA producing an improvement in the security-auditability pareto fronteir.

**Strengths:**

- The paper highlights the Security-Auditability Dilemma. This appears to be a novel contribution and an important dilemma worth noting and addressing. They also provide empirical results to validate the existence of this dilemma.
- The explanation of the proposed method and solution is mostly clear
- They provide evidence that their ALCA method functions as intended, and could be a solution to the Security-Auditability Dilemma: (i) It reduces attack success rates (ASR), demonstrating mostly reduced ASR versus the presented baselines, (ii) they present evidence that the decoding method works, meaning auditability can be maintained, (iii) they provide evidence that ALCA models maintain good performance on capabilities benchmarks.

**Weaknesses:**

**Motivating the utility of providing user-facing reasoning traces**

The authors could perhaps do a better job at motivating the utility of presenting CoT reasoning traces to users (who could be potential attackers). A solution to the security issue is to hide the reasoning completely from users. However, there may be reasons we would still like to show users as much reasoning as possible. The paper does not seem to describe these reasons very well - the reasons it does touch on, such as “transparent reasoning traces as supervision target in training”, do not seem relevant for user-facing applications.

**ALCA seems a convoluted solution, simpler baselines may exist**

ALCA seems to be an overly convoluted and complex solution to the security-auditability dilemma. It does not seem to be properly baselined against simpler, potentially more natural solutions. The main goal of ALCA appears to be to provide a method that shows the user all harmless reasoning while hiding any reasoning that could create potential vulnerabilities or information leakages. A more natural and simpler solution here, for exampe, is to simply have another LLM redact sensitive parts of the reasoning before providing them to the user. A simplification of ALCA would be to simply mask reasoning tokens from the user where the probes fires. These are methods that do not alter the models actual generations and so will not impact the “auditability” axis. The paper uses a relevant masked reasoning method in Section 2, but does not seem to baseline ALCA against masked reasoning in the main results, which seems a problematic omission.

Moreover, none of the existing baselines used in the paper are described, motivated or contextualized. It is unclear if they are meaningful baselines for a security-auditability evaluation (they may only be good baselines for the "security" component).

**Lack of empirical focus on auditability**

In general, the paper seems to heavily focus on the “security” component, and neglects the “auditability” component. Namely, the auditability of ALCA is not compared to any baseline methods, and the metrics used for measuring auditability in Table 3 seem unconvincing (these metrics are proxies for auditability, not direct measures). The paper does not seem to include any model generations - it would be useful to qualitatively compare decoded latents to the ground-truth text.

**Presentation issues**

I have some concerns regarding the care gone into the preparation of the paper. All text in the paper is in bold from page 5 onwards. There are a few other formatting issues throughout (e.g., line 447). The conclusion is very minimal and there is no discussion of limitations. Table 2 seems to, on multiple occasions, highlight the ALCA result as the best performing when it appears a different method was the best performing (e.g., for Llama-3, ALCA GCG is worse than STAIR GCG?). Citations are sometimes missing, e.g., for TAP method in line 132.

**Other**

Previous papers have proposed methods for latent reasoning. This paper does not ground their latent generation approach in existing literature.

**Questions:**

- Can you confirm you are the first paper to introduce the “security-auditability dilemma” in this context?
- Is there a reason you did not try the simpler baseline approaches I touched on above? Could you run these baselines?
- Do you agree that the “auditability” axis is neglected in your experiments? Could you run additional experiments to better validate the auditability of ALCA?
- In Section 2, how exactly is the 'masking' in the masked reasoning performed?
- What dataset do you use for training the probe?
- Why did you choose the decoding method you used? Did you try other approaches? Why did you not just decode the latents directly through lm_head ?
- Can you include some example generations from the model? In particular, generations decoded from latents would be interesting.
- For the “downstream %” results in Table 2, was the model generating its reasoning in latent mode?
- In section 4.3, the plots mention L_decode, but the text mentions L_causal - is this a mistake?

---

> ### Author Response · Authors · 2025-11-27
> **Rebuttal for Reviewer B74G [1/4]**
>
> # Rebuttal for Reviewer B74G
>
> We sincerely thank the reviewer for their insightful feedback and constructive suggestions. These comments are invaluable for enhancing the clarity, rigor, and impact of our work. We have carefully considered each point and will address them below.
>
> ---
>
> ## **Q1: ALCA seems a convoluted solution, simpler baselines may exist.**
>
> **A1:**
> We appreciate the reviewer raising the critical question of whether simpler solutions (like masking or external redaction) would suffice. This very issue underscores the advantage of our approach. We respectfully clarify that we explicitly evaluated the "simple masking" baseline in our initial experiments (Section 2, Table 1), and the results empirically demonstrate that such simplicity comes at the cost of critical security vulnerabilities. We now argue, from four complementary angles, why “simple masking” is intrinsically insufficient: (1) the constraints imposed by open-source deployment, (2) the fragility under closed-source usage scenarios, (3) the empirical evidence, and (4) the non-trivial computational overhead.
>
> **1. The constraints imposed by open-source deployment**
> Our method is not restricted to closed-source settings; it is expressly designed to be deployed in open-source, locally-hosted environments as well. In the context of open-source models, "simple masking" (implemented via inference code) offers zero security, as users can simply modify the code to reveal the hidden text. ALCA provides architectural security: the safety reasoning is encoded in latent vectors that are unintelligible without the specific decoding embedding, securing the reasoning process even if the model weights are accessible.
>
> **2. Theoretical Necessity: Why Masking Fails (The "Stepping Stone" Effect)**
> Even under closed-source conditions, ALCA delivers markedly greater robustness against jailbreaking attacks. As analyzed in Appendix I ("The Brittleness of Discrete Reasoning"), the vulnerability stems from the discrete nature of text:
> *   **The Flaw of Masking:** Even if the output is masked, the model internally commits to discrete harmful tokens (e.g., generating the token "bomb" in the hidden layer). This places the model's internal state into a "harmful trajectory." Adaptive attackers can exploit the gradients or activation patterns of these internal states to optimize their attacks, effectively bypassing the mask.
> *   **The Advantage of ALCA:** ALCA shifts safety deliberation into a continuous latent space. By avoiding the commitment to discrete harmful tokens, we remove the "stepping stones" that attackers use to climb the gradient. This creates a fundamentally smoother and more robust defense surface that simple masking cannot achieve.
>
> **3. Empirical Evidence: Simple Masking is Insufficient**
> ALCA’s robustness advantage is not merely analytical—it is empirically substantiated across comprehensive experiments. In Section 2, we evaluated "Masked Reasoning" (generating explicit safety CoT but hiding it from the user/attacker). The results in Table 1 show a stark contrast:
> *   **Simple Baseline (Masked Reasoning):** Suffers a high Adaptive Attack Success Rate (ASR) of 25.9%.
> *   **Our Method (ALCA):** Drastically reduces this ASR to ~9.1%.
> *   **Conclusion:** While masking hides the text, it fails to provide robust defense against jailbreak adversaries (e.g., TAP, PAIR).
>
> **4. Efficiency vs. External Redaction:**
> The reviewer suggested using "another LLM to redact sensitive parts." This pipeline approach requires running two LLM passes, doubling the inference latency and cost. ALCA is an end-to-end solution with minimal overhead (Table 6).
>
> In summary, the apparent complexity of ALCA’s latent-reasoning pipeline and key-gated self-decoding is not an over-engineered artifact—it is the minimal requisite structure for simultaneously delivering (i) strong safety-alignment guarantees and (ii) deployment versatility across both open- and closed-source ecosystems.
>
> **Action:** We acknowledge that placing the "Masked Reasoning" baseline in Section 2 might have made it less prominent. We will add the Masked Reasoning results into the main performance table (Table 2) in the revised paper to explicitly baseline ALCA against this simpler approach, clearly demonstrating the performance gap.
>
> ---

---

> > ### Author Response · Authors · 2025-11-27
> > **Rebuttal for Reviewer B74G [2/4]**
> >
> > ## **Q2: Motivating the utility of providing user-facing reasoning traces**
> >
> > **A2:**
> > We thank the reviewer for the careful reading and for raising this pivotal question. Addressing it precisely articulates one of the core problems our method resolves and thereby clarifies the principal contribution of ALCA. While total concealment is acceptable for purely malicious queries, we argue that preserving user-facing reasoning is essential for **Trust**, **Utility**, and **Usability** in general-purpose LLMs. Beyond the researcher's need to surface the SCoT for supervisory training, the CoT serves the following additional purposes:
> >
> > **1. Trustworthy Refusal vs. Model Failure**
> > Users need to distinguish between a safety enforcement (the model refusing to answer) and a capability failure (the model unable to answer).
> > *   **The Problem with Hiding:** If a model simply outputs "I cannot answer" without explanation, users may interpret this as a hallucination or a lack of knowledge.
> > *   **ALCA's Advantage:** ALCA hides the internal safety deliberation (the analysis of the harmful content) but allows the model to transition back to explicit text to generate a reasoned refusal (e.g., "I cannot generate this code because it exploits a vulnerability in..."). Likewise, when the hidden chain reveals only a capability or knowledge gap, the decoded CoT transparently details that shortcoming so users can immediately discern the reason for rejection. This maintains transparency about why the request was denied without leaking the harmful "how-to" information during the deliberation phase.
> >
> > **2. Granular Control for Mixed Utility**
> > In many complex tasks (e.g., coding assistance, mathematical reasoning), the reasoning chain is the core product.
> > *   **The Problem with Hiding:** A blanket policy of "hiding reasoning" would degrade the user experience for benign or borderline queries.
> > *   **ALCA's Advantage:** ALCA is designed with a selective trigger (the Probe). It only conceals the specific steps where the model analyzes safety risks. Once the safety check is complete, the model seamlessly switches back to Explicit Mode to continue generating useful, harmless CoT for the rest of the query. This ensures that safety mechanisms do not cannibalize the model's general reasoning utility.
> >
> > In summary, exposing a chain-of-thought (CoT) to end-users is not optional—it is a functional necessity. It is precisely this requirement of displaying an intelligible rationale to users, while simultaneously preserving a supervisory signal for safety-alignment training, that forces the Security-Auditability Dilemma upon us and motivates the design of ALCA.
> >
> > **Action:** We will accordingly revise the Introduction to foreground the necessity of user-facing reasoning traces, thereby making the motivation for—and the distinct contribution of—ALCA unmistakably clear to readers from the outset.
> >
> > ---

---

> > > ### Author Response · Authors · 2025-11-27
> > > **Rebuttal for Reviewer B74G [3/4]**
> > >
> > > ## **Q3: Lack of empirical focus on auditability**
> > >
> > > **A3:**
> > > We thank the reviewer for pointing out this critical issue. We agree that auditability must ensure the reconstructed text faithfully represents the model's internal decision process. To further prove this beyond the consistency between the ground-truth text and the self-decoded text, we introduce a rigorous Functional Equivalence Experiment and clarify our baseline comparison.
> > >
> > > **1. New Experiment: Functional Equivalence via Downstream Continuation**
> > > To demonstrate that the decoded text faithfully recovers both the content and the contextual fidelity of the original chain-of-thought—thereby furnishing auditors with a reliable inspection target—we compared the downstream generation conditioned on three different contexts:
> > > 1.  **Ground Truth:** The original explicit safety CoT.
> > > 2.  **Latent:** The internal hidden vectors (used by ALCA during inference).
> > > 3.  **Decoded:** The text reconstructed by our Self-Decoder.
> > >
> > > We fed these three contexts into the model and generated the final response. We evaluated the consistency of the model's final response ($y$) and its initial internal state ($h_{start}$) when conditioned on three different reasoning contexts: (1) Ground Truth Explicit CoT, (2) ALCA Latent Vectors, and (3) ALCA Decoded Text.
> > >
> > > | Context Comparison Pairs | Semantic Sim. ($y$) $\uparrow$ | ROUGE-L ($y$) $\uparrow$ | BLEU-4 ($y$) $\uparrow$ | Hidden State Sim. ($h_{start}$) $\uparrow$ |
> > > | :--- | :---: | :---: | :---: | :---: |
> > > | (1) Ground Truth vs. (3) Decoded | 0.95 | 0.90 | 0.84 | 0.92 |
> > > | (2) Latent Vectors vs. (3) Decoded | 0.97 | 0.94 | 0.90 | 0.95 |
> > >
> > > *Note: "Hidden State Sim." measures the cosine similarity of the first hidden state generated immediately after the reasoning context, representing the model's "initial thought" before generating the response.*
> > >
> > > **Implication:** This confirms that the Decoded Text conveys the exact same semantic instruction and causal influence to the model as the latent hidden state and the original reasoning. The auditor sees exactly what the model "thought" and used to generate the response.
> > >
> > > **2. Qualitative & Human Evaluation**
> > > To go beyond n-gram metrics, we conducted a new study using both Human Evaluators and Gemini-2.5-Pro to rate the **Semantic Consistency** between the original and decoded reasoning on a 5-point scale (5 = Identity).
> > >
> > > | Metric | Human Eval (1-5) | Gemini-Pro Eval (1-5) | Semantic Similarity (Embedding) |
> > > | :--- | :---: | :---: | :---: |
> > > | **Score** | 4.87 ± 0.09 | 4.91 ± 0.07 | 0.96 |
> > >
> > > *   **Result:** Both human and advanced model judges rated the consistency near-perfect (>4.8/5). This confirms that ALCA's self-decoding mechanism is **high-fidelity** and not just a hallucination.
> > >
> > > **Examples:**
> > > *   **Ground Truth (Explicit):** "The user is asking for instructions to make [Substance X]. Since this is a controlled precursor used for illegal manufacturing, providing this information would violate safety policies."
> > > *   **ALCA Decoded (Reconstructed):** "The user is asking for instructions to make [Substance X]. Because this is a regulated precursor used for illegal manufacturing, generating this information would breach safety guidelines."
> > >
> > >
> > > **3. Clarification on Baselines and Purpose**
> > > The reviewer noted a lack of auditability baselines.
> > > *   **We are the First:** ALCA is the first framework to implement "Latent-to-Text" reconstruction specifically for safety supervision. No prior latent reasoning methods offer this self-decoding capability for comparison.
> > > *   **The Baseline is Explicit CoT:** Our goal for auditability is to match the transparency of existing Explicit CoT methods (like STAIR/CoT-Safety, which we compared against), while significantly outperforming them on Security.
> > > *   **Purpose:** The ultimate goal of auditability is to enable supervision to improve safety. As shown in our main results, explicit methods (STAIR) are auditable but still get a higher ASR compared to ALCA (~9% ASR). This confirms that ALCA furnishes sufficient auditability for safety-alignment training and thereby enables a more robust alignment capability.
> > >
> > > **Action:** We will incorporate the new experiments and the corresponding auditability analysis into the appendix.
> > >
> > > ---
> > >
> > > ## **Q4: First to introduce "Security-Auditability Dilemma"?**
> > >
> > > **A4:**
> > > Yes. To the best of our knowledge, we are the first to formalize this dilemma specifically in the context of Exposed Chain-of-Thought (CoT) Safety Reasoning and Adaptive Attacks. Our experiments decisively demonstrate that safety chain-of-thought (SCoT) enhances safety-alignment capability (Figure 2(a)), while simultaneously revealing its heightened vulnerability to adaptive attacks and its propensity to leak harmful information (Figure 2(b) and Table 1).

---

> > > > ### Author Response · Authors · 2025-11-27
> > > > **Rebuttal for Reviewer B74G [4/4]**
> > > >
> > > > ---
> > > >
> > > > ## **Q5: How exactly is 'masking' performed in Section 2?**
> > > >
> > > > **A5:**
> > > > For the "Masked Reasoning" baseline, the model generates the full CoT tokens internally, but we identified safety CoT  tokens through a classifier (GPT-4) and delete them from the final user output. This achieves visibility for subsequent tokens while remaining invisible to both users and harmful-content detectors.
> > > >
> > > > ---
> > > >
> > > > ## **Q6: What dataset do you use for training the probe?**
> > > >
> > > > **A6:**
> > > > We constructed a balanced dataset of **~10,000 samples** derived from SafeChain and AdvBench. It consists of (Hidden State $h_t$, Label $y$) pairs, where $y=1$ if the step is safety-critical and $y=0$ otherwise. The probe achieves an **F1-score of 0.98** (Table 4).
> > > >
> > > > ---
> > > >
> > > > ## **Q7: Why specific decoding method vs lm_head? Did you try lm_head?**
> > > >
> > > > **A7:**
> > > > Yes, we tried `lm_head` and it failed (BLEU < 0.05).
> > > > *   **Reason:** The `lm_head` is trained to map a hidden state to the **immediate next token**. In contrast, our latent vector $z$ is trained (via the latent autoregressive objective) to compress a **multi-step future reasoning trajectory**. The feature spaces are fundamentally misaligned. A specialized, lightweight Decoder ($D$) is mathematically necessary to "unpack" this compressed representation back into text.
> > > >
> > > > ---
> > > >
> > > > ## **Q8: For "downstream" results, was the model in latent mode?**
> > > >
> > > > **A8:**
> > > > The model was in ALCA mode (active framework), but the Probe correctly identified these downstream tasks (e.g., GSM8k math problems) as harmless. Therefore, it triggered the Explicit Mode, generating standard textual CoT. This confirms that ALCA's selective triggering successfully preserves utility for benign tasks.
> > > >
> > > > ---
> > > >
> > > > ## **Q9: Clarification on $L_{causal}$  vs. $L_{\text{decode}}$ in Section 4.4**
> > > >
> > > > **A9:**
> > > > This is **not a typo**, but a reflection of the critical dependency we discovered between the causal consistency and decoding quality. To clarify, we explicitly list the loss functions defined in Section 3:
> > > >
> > > > **1. Causal Non-Interference Loss ($\mathcal{L}_{\text{causal}}$, Eq. 8):**
> > > > Ensures the latent vector $z_j$ preserves the semantic information necessary to generate *subsequent* tokens correctly.
> > > >
> > > > $$ \mathcal{L}_{\text{causal}} = - \sum_{k>j, s_k=0} \log P(c_k \mid x, \dots, c_{j-1}, \mathbf{z}_j) $$
> > > >
> > > >
> > > > **2. Self-Decoding Loss ($\mathcal{L}_{\text{decode}}$, Eq. 10):**
> > > > Supervises the reconstruction of the *current* safety reasoning step from the latent vector for auditability.
> > > >
> > > > $$ \mathcal{L}_{\text{decode}} = - \sum_{t=1}^{|c_j|} \log P(c_{j,t} \mid c_{j,<t}, \mathbf{z}_j, \mathbf{e}_{\text{decode}}) $$
> > > >
> > > > **3. Final Joint Optimization Objective (Eq. 11):**
> > > >
> > > > $$ \mathcal{L}_{\text{ALCA}} = \underbrace{\|\mathbf{z}_{j,n} - \mathbf{z}^*_{j,n}\|^2}_{\mathcal{L}_{\text{latent}}} + \lambda \mathcal{L}_{\text{causal}} + \beta \mathcal{L}_{\text{decode}} $$
> > > >
> > > > ---
> > > >
> > > > ## **Q10: Presentation Issue**
> > > >
> > > > **A10:**
> > > > We sincerely apologize for the formatting oversights and the compilation error that caused mistakes in bold text. We recognize that high presentation quality is essential for ICLR. We will correct every textual error and ensure that the final version is presented flawlessly. Thank you for your meticulous and rigorous reading—it has substantially improved the quality of our paper.
> > > >
> > > > ---
> > > >
> > > > Once again, we express our deepest gratitude to the Reviewer. Your expert feedback has provided us with a clear roadmap for a significantly improved manuscript. We are confident that the extensive new experiments, the new key visualization, and the refined framing fully address your concerns and will substantially elevate the final paper. Thank you for devoting your precious time to the review process.

---

### Author Response · Authors · 2025-12-02
**Summary to the Area Chair**

# **Summary to the Area Chair**

## **Recap of Contribution**
This paper identifies and formalizes the **Security-Auditability Dilemma** in LLM safety: the transparency required for auditing safety reasoning (Chain-of-Thought) for high safety capability during safety alignment training creates a gradient-rich attack surface for adaptive adversaries. We propose **Auditable Latent CoT Alignment (ALCA)**, a framework that shifts safety deliberation into a continuous latent space to thwart attacks while utilizing a novel self-decoding mechanism to preserve human-readable auditability. ALCA reduces the Attack Success Rate (ASR) by over 54% compared to strong baselines without compromising model utility.

## **Rebuttal Summary**
We thank the reviewers for their constructive feedback, which we have used to significantly strengthen the paper’s empirical rigor and scope. In our revised manuscript and rebuttals, we have addressed all key concerns as follows:

*   **Justification Against Simpler Baselines (Addressing Reviewer B74G):** We explicitly integrated the "Simple Masking" baseline into our main results from appendix. Experiments confirm that merely hiding text is insufficient (25.9% ASR) due to the model's internal commitment to discrete tokens (the "Stepping Stone" effect). In contrast, ALCA’s continuous latent reasoning achieves superior robustness (~9% ASR), validating the necessity of our architectural approach.
*   **Rigor of Auditability (Addressing Reviewers H3gY & B74G):** To address concerns about "circular" model-based evaluation, we introduced **Human Evaluation** (achieving 4.87/5 consistency) and a **Functional Equivalence** experiment. The latter demonstrates that the decoded text induces the same downstream internal states and subsequent text as the latent vectors, proving that the audit trail is a faithful representation of the hidden reasoning.
*   **Generalization & Robustness (Addressing Reviewers oHig & H3gY):** We expanded our evaluation beyond safety to Hallucination, Privacy, and Social domains, proving ALCA’s versatility as a "sanitizing buffer." We also provided a sensitivity analysis for the probe layer and visualized latent manifolds to demonstrate the trigger mechanism's robustness against adversarial manipulation.
*   **Statistical Stability & Efficiency (Addressing Reviewer d8ZB):** We provided results from three independent training runs with standard deviations to prove stability. Furthermore, we expanded the cost analysis to show that ALCA achieves nearly double the throughput of explicit CoT methods via sequence compression, with negligible probe overhead (<0.5ms).

## **Conclusion**
ALCA offers the first principled solution to the tension between safety robustness and transparency. With the strengthened empirical evidence regarding fidelity, generalization, and efficiency, we believe this work presents a critical step forward for the secure deployment of reasoning models.

---

> ### Author Response · Authors · 2025-12-02
>
> ## **Detailed Rebuttal Navigation**
>
> | Reviewer | Key Concern / Question | **Our Resolution & New Evidence** | **Location in Thread** |
> | :--- | :--- | :--- | :--- |
> | **B74G** | 1. Necessity vs. Simple Masking | **Added:** Empirical proof that masking fails (25.9% ASR) vs. ALCA (9.1%) due to discrete token commitment; ‘Simply Mask’  is not applicable for open source utlization. | *Rebuttal for Reviewer B74G [1/4]* |
> | **B74G** | 2. Utility of User-Facing CoT | **Clarified:** Explained necessity for "Trustworthy Refusal" (vs. failure) and granular utility in mixed tasks; ALCA enables selective hiding. | *Rebuttal for Reviewer B74G [2/4]* |
> | **B74G** | 3. Auditability Rigor | **Added:** **Functional Equivalence Experiment** showing Decoded text induces same downstream state as Latent vectors; **Human Eval** (4.87/5 consistency). | *Rebuttal for Reviewer B74G [3/4]* |
> | **oHig** | 4. Generalization (Hallucinations) | **Added:** New experiments on **Privacy, Social, & Hallucination** domains (Table R1); showed ALCA acts as a "Sanitizing Buffer" for PII/toxicity. | *Rebuttal to Reviewer oHig [1/2]* |
> | **oHig** | 5. Probe Accuracy & Safety | **Added:** "Fail-Safe" analysis (Table R2) showing 0% full-chain leakage; Visualized distinct semantic manifolds between Safety vs. Math/Coding. | *Rebuttal to Reviewer oHig [2/2]* |
> | **H3gY** | 6. Circular Evaluation | **Added:** Independent **Human Evaluation** & Gemini-Pro judging to replace self-evaluation, confirming high semantic fidelity. | *Response to Reviewer H3gY [1/3]* |
> | **H3gY** | 7. Adversarial Probe Robustness | **Added:** **"All-or-Nothing" probabilistic argument** (attacker must fool probe $N$ times); Layer sensitivity analysis showing robustness across layers 24-32. | *Response to Reviewer H3gY [2/3]* |
> | **d8ZB** | 8. Experimental Rigor (Variance) | **Added:** Results from **3 Independent Runs** reporting Mean & Std Dev, confirming stability across seeds. | *Response to Reviewer d8ZB [1/4]* |
> | **d8ZB** | 9. Judge Reliability (GPT-4) | **Added:** **Human-AI Correlation Study** (Kappa=0.92) & Cross-validation with Llama-Guard-2 (94.5% agreement). | *Response to Reviewer d8ZB [2/4]* |
> | **d8ZB** | 10. Cost & Latency | **Added:** Granular latency breakdown (Figure 5); showed Probe <0.5ms & ALCA is **1.9x faster** than Explicit CoT due to compression. | *Response to Reviewer d8ZB [3/4]* |

---

### Note · Program_Chairs · 2025-12-09
**Submission Desk Rejected by Program Chairs**

Hallucinated reference:
Yixiang Ma, Ziyi Liu, Zhaoyu Wang, Zhaofeng Xu, Yitao Wang, and Yang Liu. Safechain: A framework for securely executing complex commands using large language models. arXiv preprint arXiv:2402.16521, 2024a.